# HMGB1 coordinates SASP-related chromatin folding and RNA homeostasis on the path to senescence

Konstantinos Sofiadis[1,†], Natasa Josipovic[1,†], Milos Nikolic[2,†] (iD), Yulia Kargapolova[2,‡] (iD), Nadine Übelmesser[1], Vassiliki Varamogianni-Mamatsi[1], Anne Zirkel[2], Ioanna Papadionysiou[1], Gary Loughran[3], James Keane[3,4] (iD), Audrey Michel[3], Eduardo G Gusmao[1], Christian Becker[5], Janine Altmüller[5], Theodore Georgomanolis[2,5] (iD), Athanasia Mizi[1] & Argyris Papantonis[1,2,*] (iD)

## Abstract

**Spatial organization and gene expression of mammalian chromosomes are maintained and regulated in conjunction with cell cycle progression. This is perturbed once cells enter senescence and the highly abundant HMGB1 protein is depleted from nuclei to act as an extracellular proinflammatory stimulus. Despite its physiological importance, we know little about the positioning of HMGB1 on chromatin and its nuclear roles. To address this, we mapped HMGB1 binding genome-wide in two primary cell lines. We integrated ChIP-seq and Hi-C with graph theory to uncover clustering of HMGB1-marked topological domains that harbor genes involved in paracrine senescence. Using simplified Cross-Linking and Immuno-Precipitation and functional tests, we show that HMGB1 is also a *bona fide* RNA-binding protein (RBP) binding hundreds of mRNAs. It presents an interactome rich in RBPs implicated in senescence regulation. The mRNAs of many of these RBPs are directly bound by HMGB1 and regulate availability of SASP-relevant transcripts. Our findings reveal a broader than hitherto assumed role for HMGB1 in coordinating chromatin folding and RNA homeostasis as part of a regulatory loop controlling cell-autonomous and paracrine senescence.**

**Keywords** 3D chromatin folding; loop; replicative senescence; RNA splicing; senescence-associated secretory phenotype
**Subject Categories** Chromatin, Transcription & Genomics; RNA Biology
**Mol Syst Biol. (2021) 17: e9760**

## Introduction

The high-mobility group box 1 (HMGB1) protein, a member of the highly conserved non-histone DNA-binding HMG protein family, was named after its characteristically rapid electrophoretic mobility (Stros, 2010). HMGB1 is the most abundant non-histone protein in mammalian nuclei, with 1 HMGB1 molecule per every ~ 10 nucleosomes (Thomas & Stott, 2012). Despite its high abundance and conservation, HMGB1 has been predominantly studied as an extracellular signaling factor, hence its characterization as an "alarmin" (Lohani & Rajeswari, 2016; Bianchi *et al*, 2017).

To function as an alarmin, HMGB1 is actively secreted by cells such as activated monocytes and macrophages or passively released by necrotic and damaged cells. Once received by other cells in the niche, HMGB1 is recognized by RAGE receptors to potently signal inflammation (Scaffidi *et al*, 2002; Bonaldi *et al*, 2003; Orlova *et al*, 2007). In cells entering senescence, HMGB1 translocates from the nucleus to the cytoplasm and is then secreted to stimulate NF-κB activity via Toll-like receptor signaling. Its relocalization and secretion control the senescence-associated secretory phenotype (SASP) of cells, thus representing a major paracrine contributor both *in vitro* and *in vivo* (Salminen *et al*, 2012; Acosta *et al*, 2013; Davalos *et al*, 2013).

Inside proliferating cell nuclei, HMGB1 has been studied in some detail for its contribution to DNA repair (Ito *et al*, 2015; Mukherjee & Vasquez, 2016), V(D)J recombination (Little *et al*, 2013; Zagelbaum *et al*, 2016) or chromatin assembly (Bonaldi *et al*, 2002), but far less for its transcriptional role (Calogero *et al*, 1999; Mitsouras *et al*, 2002). Cells lacking HMGB1 contain reduced numbers of nucleosomes, rendering chromatin more susceptible to DNA damage, spurious transcription, and inflammatory activation (Giavara *et al*, 2005;

1 Institute of Pathology, University Medical Center Göttingen, Göttingen, Germany
2 Center for Molecular Medicine Cologne, University of Cologne, Cologne, Germany
3 Ribomaps, Cork, Ireland
4 Cork Institute of Technology, Cork, Ireland
5 Cologne Center for Genomics, University of Cologne, Cologne, Germany
*Corresponding author. Tel: +49 551 39 65734; E-mail: argyris.papantonis@med.uni-goettingen.de
†These authors contributed equally to this work
‡Present address: Heart Center, University Hospital Cologne, Cologne, Germany

El Gazzar *et al*, 2009; Celona *et al*, 2011; De Toma *et al*, 2014). As regards its association with chromatin, HMGB1 is thought to bind it in a nonspecific manner via its two HMGB-box domains. This allows it to bend and contort DNA and, thus, facilitate recruitment of transcription factors such as p53 (Stros, 2010; Rowell *et al*, 2012). HMGB1 associates with cognate DNA sites via characteristically high "on/off" rates, while its acidic tail is important for stabilizing binding (Pallier *et al*, 2003; Ueda *et al*, 2004; Stros, 2010; Blair *et al*, 2016). However, HMG-box DNA-binding domains are particularly insensitive to standard fixatives such as formaldehyde (Pallier *et al*, 2003; Teves *et al*, 2016). Thus, capturing HMGBs on chromatin is challenging, and there exist no datasets describing HMGB1 binding in mammalian cells (see http://chip-atlas.org). As a result, our appreciation of its on-chromatin roles remains vague.

To address this, we employed a tailored approach that previously allowed us to efficiently map HMGB2 binding genome-wide (Zirkel *et al*, 2018). We can now show that HMGB1 binding in primary endothelial and lung fibroblast cells is far from nonspecific, while also disparate to that by HMGB2. Following integration of its binding positions with genome-wide chromosome conformation capture data (Hi-C), we found that HMGB1 demarcates the boundaries of a subset of topologically associating domains (TADs; Dixon *et al*, 2012; Nora *et al*, 2017) and loop domains (Rao *et al*, 2014). This topological contribution is eliminated upon senescence entry, and knockdown/overexpression experiments show that HMGB1 controls the expression of genes that are central to the senescent program and embedded in these domains. Critically, as HMGB1 was proposed to have RNA-binding capacity (Castello *et al*, 2016), we used simplified Cross-Linking and Immuno-Precipitation (sCLIP) (Kargapolova *et al*, 2017) to show it also influences the availability of senescence-relevant mRNAs. This occurs via a network of RNA-binding factors interacting with HMGB1. In summary, using replicative senescence as a model, we characterize the multiple roles of HMGB1 that converge on the coordination of chromatin and RNA control for SASP regulation.

# Results

### Senescence entry is marked by HMGB1 nuclear loss and secretion

To investigate the nuclear roles of HMGB1 across cellular contexts, we used primary human umbilical vein endothelial cells (HUVECs) and fetal lung fibroblasts (IMR90) that are of distinct developmental origin and have disparate gene expression programs. We defined an early-passage (proliferative) and a late-passage (senescent) state by combining β-galactosidase staining, cell cycle staging by FACS, and MTT proliferation assays (Fig 1A), as well as a "senescence clock" based on the methylation of six CpGs (Zirkel *et al*, 2018). Next, we used RNA-seq data from proliferating and senescent HUVEC and IMR90 (from two different donors/isolates) to look into changing mRNA levels of chromatin factors. Significant and convergent changes between the two cell types involved strong suppression of histone chaperones, heterochromatin-, polycomb-, and lamina-associated proteins, centromere components, cohesin, and condensin complexes, as well as all HMGB/N-family proteins (Fig 1B). Many of these factors were consistently suppressed also at the protein level (Fig 1C; Davalos *et al*, 2013; Shah *et al*, 2013; Rai *et al*, 2014;

Zirkel *et al*, 2018). We focused on HMGB1 due to its conservation, high nuclear abundance (Thomas & Stott, 2012, Fig 1D), and key role in SASP induction (Davalos *et al*, 2013), but mostly due to its elusive role on chromatin, especially in respect to spatial chromosome organization.

Immunodetection in early- and late-passage cells documented a > 50% decrease in HMGB1 nuclear levels in the heterogeneous senescence entry populations of HUVEC or IMR90 (Fig 1E). HMGB1 nuclear depletion was most dramatic in the enlarged senescent nuclei of either cell type, while smaller nuclei remained largely unaffected. FACS-sorting IMR90 based on light scattering allowed enrichment for cell populations with enlarged nuclei (i.e., ~ 70% of cells had larger than average nuclei, with > 35% being > 1.5-fold larger; Appendix Fig S1A). This showed that enlarged nuclei lacking HMGB1 almost invariably represent senescent cells and harbor reduced levels of H3K27me3, a mark of facultative heterochromatin —an effect which would otherwise be masked (Fig 1C and Appendix Fig S1B and C). Last, we showed that it is these larger cells that secrete HMGB1, but not HMGB2, into the growth medium to presumably contribute to paracrine senescence (Appendix Fig S1D).

### The senescence program is predominantly transcriptionally driven

Despite strong changes documented by RNA-seq, it is still not known to which extent the senescent program is implemented via changes at the transcriptional or the translational level. To address this, we generated matching mRNA-seq, Ribo-seq, and whole-cell proteomic data from proliferating and senescent IMR90 in biological triplicates. Comparative analysis of mRNA-seq and Ribo-seq data showed that essentially all significant changes at the level of mRNA translation are matched by equivalent changes in transcript availability (Fig 1F). Few transcripts (~ 800) showed increased translation that counteracted transcriptional suppression (e.g., *TNFRSF19*, *HMGN2*, *LMNB2*) or the converse (e.g., *IL12A*, *CDKN1A*, *CDKN2B*). Gene set enrichment analysis of these two subgroups (with a "buffer" ratio of at least $\log_2 0.6$) showed that mRNAs translationally upregulated while transcriptionally suppressed are linked to the formation and secretion of endosomal vesicles (and thus to HMGB1 release into the extracellular milieu; Appendix Fig S1E), as well as to cell cycle regulation. On the other hand, transcripts translationally downregulated but transcriptionally upregulated associate with ribosome complex formation and translation, and also with RNA binding and mRNA catabolism (Appendix Fig S1F).

Similar analysis of mRNA-seq against whole-cell proteome data verified that the vast majority of proteins with significantly altered levels in senescence were similarly regulated transcriptionally (Fig 1G). Genes linked to the hallmark GO terms and pathways of senescence entry were convergently up- (e.g., ECM organization, lysosome) or downregulated at both the mRNA and protein levels (e.g., cell cycle, DNA conformation change, RNA processing; Fig 1G and H). Curiously, several processes relevant to the SASP (e.g., TNFα/NF-κB signaling) appeared regulated by a combination of higher transcription and diminished protein availability (Fig 1H and Appendix Fig S1G). Our findings correlate with Ribo-seq analyses in a model of oncogene-induced senescence (Loayza-Puch *et al*, 2013), which also highlighted that cell cycle arrest is mainly driven by transcriptional changes.

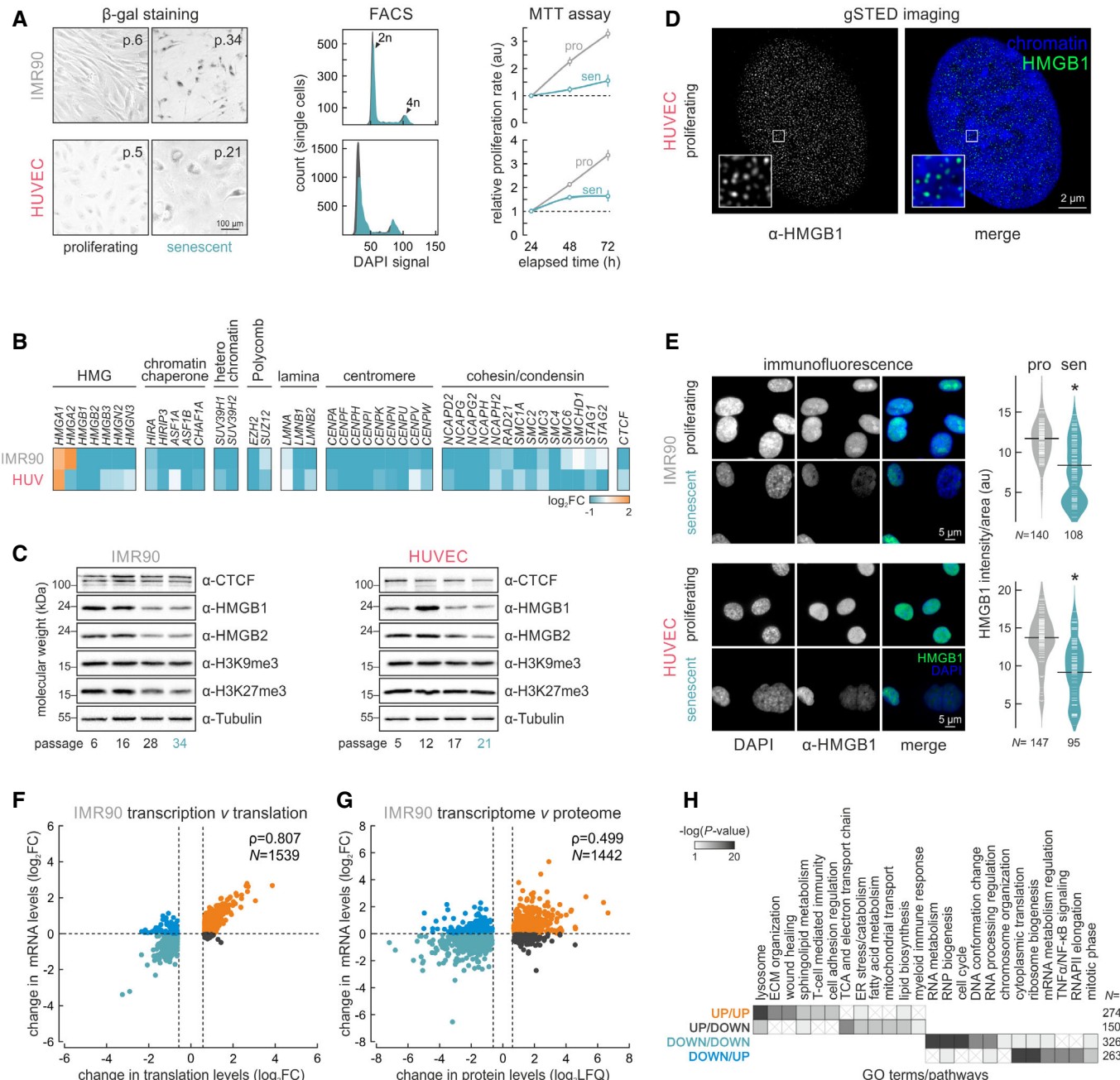

**Figure 1. Senescence entry by human cells is mostly transcriptionally driven.**

A   Proliferating and senescent IMR90 and HUVECs assayed for β-galactosidase activity (*left*), cell cycle profiling via FACS (*middle*), and proliferation via MTT assays (*right*).

B   Heatmaps showing changes in gene expression levels upon senescence (log₂FC) of genes encoding selected chromatin-associated factors. For each gene shown, statistically significant expression changes were recorded in at least one cell type.

C   Western blots showing changing protein levels on the path to senescence in IMR90 and HUVECs. Passage 6 cells represent the proliferating state, and passages 21 and 34 the senescent state for HUVECs and IMR90, respectively.

D   Super-resolution (gSTED) imaging of HMGB1 distribution in proliferating HUVEC nuclei counterstained with DAPI. Bar: 2 μm.

E   Representative immunofluorescence images of IMR90 and HUVECs (*left*) showing reduced HMGB1 levels in senescent nuclei; bean plots quantify this reduction (*right*; N is the number of cells analyzed per condition/cell type). Bars: 5 μm. *P < 0.01; Wilcoxon–Mann–Whitney test.

F   Scatter plots showing correlation between RNA-seq (transcription) and Ribo-seq (translation; log₂) in proliferating and senescent IMR90. Pearson's correlation values (ρ) and the number of genes in each plot (N) are also shown.

G   As in panel F, but correlating RNA-seq and whole-cell proteome changes.

H   Heatmap showing GO terms/pathways associated with the gene subgroups from panel G (color-coded the same way). The number of genes in each subgroup (N) is indicated.

## HMGB1 binds active chromatin loci in a cell type-specific manner

Capturing HMGB proteins on chromatin has proven challenging, because their HMG-box DNA-binding domains are not compatible with standard formaldehyde fixation (Pallier *et al*, 2003; Teves *et al*, 2016). Here, we employed a tailored dual-cross-linking ChIP strategy to efficiently capture HMGB1 bound to its cognate sites genomewide in both proliferating HUVECs and IMR90 (Fig 2A and Appendix Fig S2A; see Materials and Methods for details). HMGB1 binding is restricted to regions marked by H3K27ac (Appendix Fig S2B), the vast majority of which are promoters and gene bodies of active genes (92% and 73% of 2038 and 1611 peaks in HUVECs and IMR90, respectively; Fig 2B). The two cell types share > 550 HMGB1 peaks, which is more than would be expected by chance, but still > 1,000 cell type-specific peaks remain that can be justified by the different gene expression programs of HUVECs and IMR90 (Fig 2C). We also used senescent IMR90 to ask whether residual HMGB1 (< 30% that of proliferating cells; Appendix Fig S2C) is redirected to particular loci. The fact that we discovered just 44 peaks in our ChIP-seq replicates argues against this possibility (Fig 2A–C).

Although HMGBs are assumed to bind chromatin in a nonspecific manner (Stros, 2010), *de novo* motif analysis in DNase I hypersensitive "footprints" (ENCODE Project Consortium, 2012) under HMGB1 peaks revealed that HMGB1 prefers similar G/A-rich motifs in IMR90 and HUVECs (Appendix Fig S2D). We also surveyed these accessible footprints for known transcription factor (TF) motifs to infer co-bound complexes or TFs that might replace HMGB1 on senescent chromatin. Of the many motifs discovered, we focused on those from factors differentially regulated upon senescence entry. In both HUVECs and IMR90, E2F-family motifs were enriched and these cell cycle-regulating TFs (e.g., E2F2 and E2F6) are strongly downregulated in senescence. In contrast, motifs of proinflammatory and senescence-activated TFs (e.g., NFKB2 and STAT1) were enriched therein and could presumably take over these positions to facilitate inflammatory stimulation and the SASP (Appendix Fig S2E).

To identify genes regulated by HMGB1, we crossed ChIP-seq with RNA-seq data in both cell types. Looking at genes directly bound by HMGB1 (i.e., excluding intergenic peaks > 10 kbp from TSSs), a first observation was that these are involved in various signaling cascades (most notably the TNFα one), as well as in cellular senescence (Appendix Fig S2F). Interestingly, HMGB1 consistently associated with more up- rather than downregulated genes in both HUVECs and IMR90. Of those, downregulated genes were involved in cell cycle transitions, while upregulated ones in ECM organization, cell adhesion, and inflammatory signaling (Appendix Fig S2G and H). Together, this analysis demonstrates that HMGB1 binds active loci relevant to the induction of the senescence gene expression program.

## HMGB1 marks a subset of senescence-regulated domain boundaries

Topologically associating domains are often considered the building blocks of chromosomes (Beagan & Phillips-Cremins, 2020), their boundaries representing sites of local insulation for spatial chromatin interactions. TAD boundaries are often marked by the presence of CTCF and/or active gene promoters (Dixon *et al*, 2012; Nora *et al*, 2017). We recently showed that a considerable number of TAD boundaries in proliferating human cells are marked by HMGB2 and that these boundaries are remodeled upon senescence entry and HMGB2 nuclear loss (Zirkel *et al*, 2018). We now reasoned that HMGB1 may also function similarly. To test this, we used Hi-C data from proliferating IMR90 and found that the majority of HMGB1 ChIP-seq signal (normalized to input) resides inside TADs (called at 40-kbp resolution; Fig 2D and E). Nevertheless, ~ 10% of HMGB1 peaks reside at TAD boundaries at positions not overlapping CTCF. Conversely, CTCF-marked TAD boundaries show no enrichment for HMGB1 (Fig 2F). Upon senescence entry, TAD boundaries that lose HMGB1 demarcation exhibit a reduction in insulation score indicative of 3D interaction reshuffling (Fig 2G), but without completely losing their insulatory character.

High-resolution Hi-C studies showed that human chromosomes are populated with sub-TAD loop domains anchored at CTCF/cohesin-bound sites (Rao *et al*, 2014). We mapped loop domains in our proliferating and senescent Hi-C data to discover that a considerable number of loops either weaken (*N* = 745) or emerge upon IMR90 senescence entry (*N* = 2,825), while 1603 loops

---

**Figure 2. HMGB1 binds active genomic loci and demarcates a subset of TAD and loop domains.**

A  Genome browser view of HMGB1 ChIP-seq (normalized to input; mean from two replicates) from proliferating HUVEC (*red*) or proliferating/senescent IMR90 (*gray/green*) in the *TMEM92* locus.

B  Bar graphs showing the genomic distribution of HMGB1 ChIP-seq peaks in HUVEC and IMR90. The number of peaks (*N*) analyzed per each cell type is indicated.

C  Venn diagram showing HMGB1 ChIP-seq peaks shared between proliferating HUVEC (*red*) or proliferating/senescent IMR90 (*gray/green*) data. *P < 0.001; more than expected by chance, hypergeometric test.

D  Exemplary Hi-C heatmap for a subregion of IMR90 chr6 aligned to HMGB1 ChIP-seq; peaks at TAD boundaries (*orange lines*) are indicated (*red arrowheads*).

E  Line plots showing normalized HMGB1 (*red*) and RAD21 ChIP-seq signal (*dark gray*) along TADs ±20 kbp from proliferating IMR90.

F  Line plots showing normalized HMGB1 (*red*) and CTCF ChIP-seq signal (*dark gray*) in the 160 kbp around HMGB1- (*left*) or CTCF-marked TAD boundaries (*right*).

G  Line plots showing mean normalized insulation scores calculated from proliferating (*black*) and senescent IMR90 Hi-C data (*blue*) in the 240 kbp around HMGB1 peaks residing at TAD boundaries (*top*) or not (*bottom*).

H  Aggregate plots showing 20 kbp resolution Hi-C signal for proliferating (*top*), senescent-specific (*bottom*), or shared loops (*middle*). The number of loops in each category (*N*) is indicated.

I  Line plots showing normalized HMGB1 (*red*) and RAD21 ChIP-seq signal (*gray*) along proliferating (*top*) or shared loop domains (*bottom*) ±50 kbp from proliferating IMR90. The percentage of HMGB1 peaks residing at loop anchors is indicated.

J  Box plot (*left*) showing significantly up/downregulated IMR90 genes harbored inside the 607 loops from panel H. The number of genes in each group (*N*) is indicated; center lines represent medians, box limits indicate the 25th and 75th percentile, whiskers extend 1.5 times the 25th–75th interquartile range, and outliers are represented by dots. Bar plots (*right*) show Go terms associated with these up/downregulated genes.

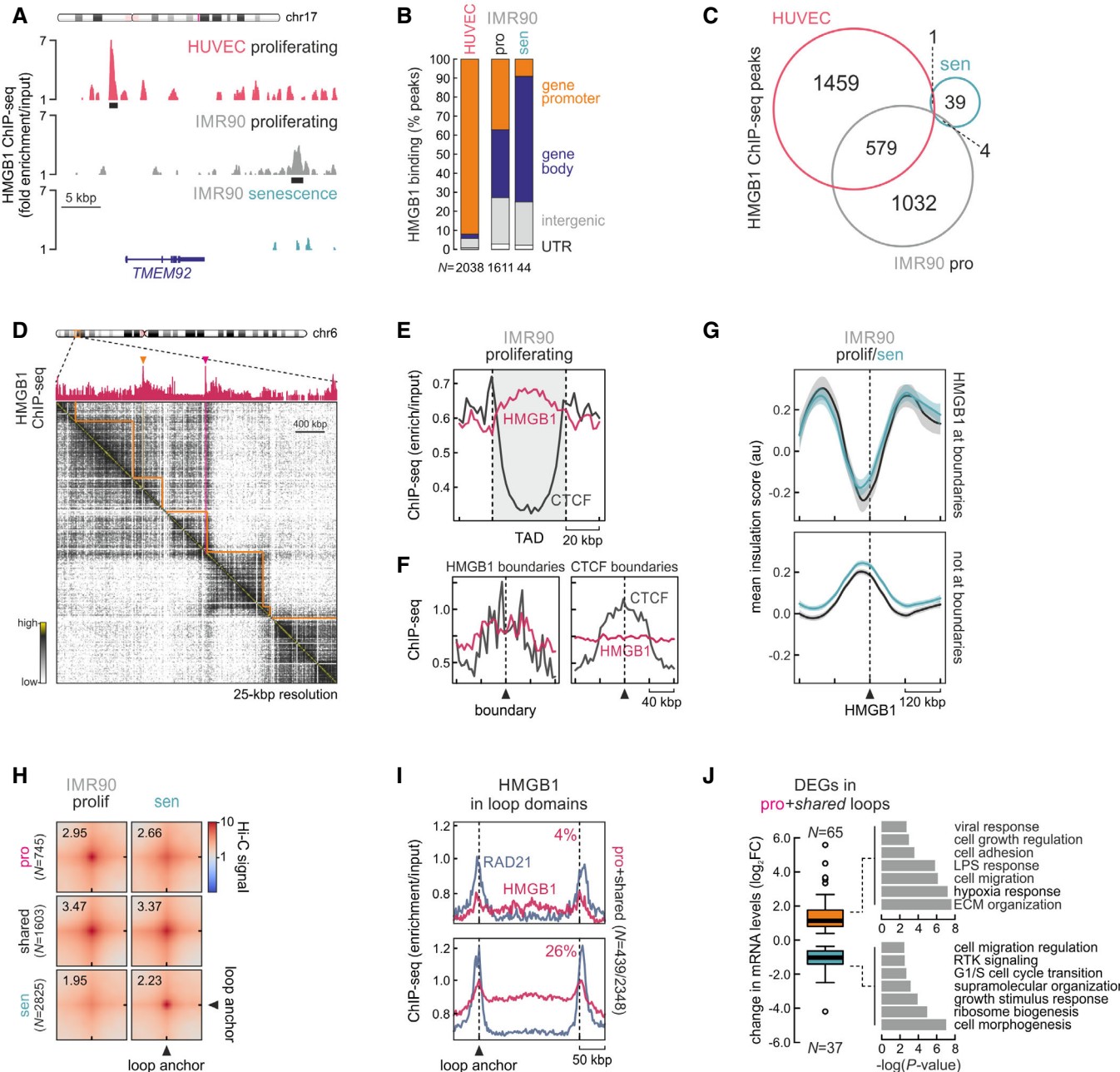

**Figure 2.**

remain invariable (Fig 2H). Remarkably, 4% of HMGB1 peaks mark the anchors of proliferating-specific loops and 26% mark invariant loops anchors; this constitutes almost 1/5 of all loop domains in proliferating IMR90 (Fig 2I). Given the loss of HMGB1 in senescence, we reasoned that genes within these loop domains would be deregulated. Indeed, these loops encompass 105 significantly regulated genes, 2/3 of which are upregulated in senescence, consistent with HMGB1-based regulation (see Appendix Fig S3A), and associate with pathways central to senescence induction, especially inflammatory activation and cell cycle arrest (Fig 2J). These effects held true also when analyzing HMGB1 ChIP-seq and Hi-C data from HUVECs (Appendix Fig S3A–F). Our analyses suggest that the

topological contribution of HMGB1 is relevant for gene regulation on the path to senescence.

## Spatial TAD co-association reveals functional specialization of chromosome domains

Given that HMGB1-marked boundaries are not fully abolished upon its senescence-induced loss, but chromosomes undergo large-scale changes upon replicative senescence entry (Zirkel *et al*, 2018), which are further accentuated in "deep" senescence (Criscione *et al*, 2016), we looked into how TADs in each chromosome associate with one another in higher-order "metaTAD"-like conformations

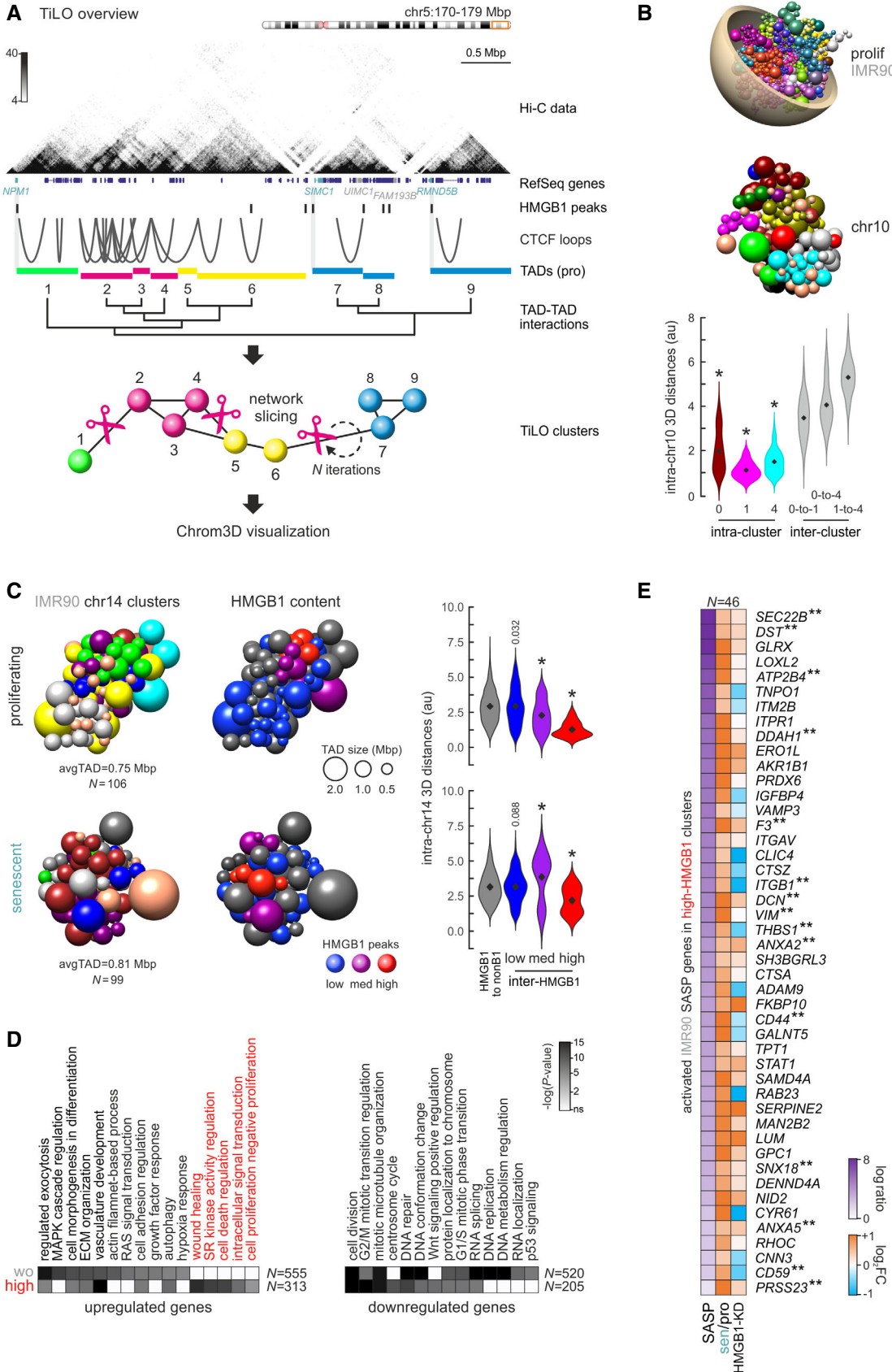

**Figure 3.**

**Figure 3.  HMGB1 marks co-associating TAD clusters harboring SASP-related genes.**

A   Overview of TiLO using a subregion of HUVEC chr5 as an example. TADs along each chromosome are treated as nodes in an interaction network, and inter-TAD Hi-C signal is used to infer network connections. Inferred connections are then sliced, and network robustness is assessed iteratively to obtain the final clustering. TAD clusters are visualized in the space of the nucleus using Chrom3D.

B   Chrom3D visualization of the whole genome (*top*) and of TAD clustering in chr10 from proliferating IMR90 (*middle*); each sphere represents one TAD. Violin plots (*bottom*) show 3D distances among TADs in three randomly selected clusters (0, 1, and 4) or between TADs from different clusters. *: significantly different to inter-cluster distances, Wilcoxon–Mann–Whitney test.

C   Chrom3D visualization of TAD clustering in chr14 from proliferating (*top*) and senescent IMR90 (*bottom*). TADs (*spheres*) are colored by the cluster they belong to (*left*) or according to their HMGB1 ChIP-seq content (*middle*; gray — zero peaks, blue — 1 or 2 peaks, purple — 3 or 4 peaks, red — 5 or more peaks). Violin plots (*right*) show 3D distances among TADs in each subgroup or between HMGB1-containing and non-containing TADs. *: significantly different to inter-cluster distances, Wilcoxon–Mann–Whitney test.

D   Heatmaps showing GO terms associated with differentially expressed genes in two TAD groups from panel C; SASP-related GO terms are highlighted. The number of genes (N) behind each heatmap are indicated.

E   Heatmap showing protein (SASP, from http://www.saspatlas.com/; log fold ratio) and gene expression levels (log$_2$FC) of IMR90 SASP-related genes embedded in high-HMGB1 (red) TADs like those in panel C. **: genes bound by HMGB1 in ChIP-seq data are more than expected by chance; $P > 0.001$, hypergeometric test.

(Fraser *et al*, 2015). We employed an unsupervised approach inspired from "topologically intrinsic lexicographic ordering" (TiLO; Johnson, 2012). Here, TADs are treated as nodes in a clustered spatial network inferred on the basis of inter-TAD Hi-C interactions that is then tested for robustness via iterative network slicing (Fig 3A; see Materials and Methods for details).

We applied TiLO to TADs derived from proliferating and senescent HUVEC and IMR90 Hi-C data and visualized the final output using Chrom3D (Paulsen *et al*, 2018) that also takes into account interchromosomal contacts and interactions with the lamina to render chromosome models (for an example see Fig 3B and Appendix Fig S4B). Although the fraction of smaller and larger TAD clusters emerging upon senescence differs between HUVECs and IMR90 (Appendix Fig S4A), there is an overall trend for larger and fewer clusters in senescent chromosomes (Fig 3C and Appendix Fig S4C and D). This is consistent with the general spatial chromatin compaction observed upon senescence entry (Criscione *et al*, 2016; Zirkel *et al*, 2018) and the fact that 43% of HUVEC and 26% of IMR90 TADs were found to merge into larger ones in Hi-C data (Zirkel *et al*, 2018; see example in Appendix Fig S4E). Our TiLO-Chrom3D combination recapitulates these organizational changes (Appendix Fig S4F).

Closer inspection of the distribution of HMGB1 peaks along TiLO clusters (Appendix Fig S4A) and of Hi-C/ChIP-seq data showing HMGB1-rich domains being depleted of CTCF loops (Fig 3A), led us to query the relative position of HMGB1-marked TiLO clusters in our models. We grouped clusters by the number of HMGB1 peaks they carry as follows: "non-HMGB1" clusters carrying zero peaks, "low" carrying 1–2, "medium" carrying 3–4, and "high" carrying 5 or more peaks; most IMR90 and HUVEC chromosomes do carry such HMGB1 "high" hot spot clusters (e.g., IMR90 chr20 does not). Strikingly, in both HUVEC and IMR90 chromosomes, clusters with increasing numbers of HMGB1 peaks are positioned progressively closer together. "High" HMGB1-hotspot clusters are closer to one another than to any other cluster (Fig 3C and Appendix Fig S4C and D). This specialized spatial clustering is reshuffled in senescent chromosomes, although former hotspot clusters remain on average closer together (Fig 3C and Appendix Fig S4C and D). To find out whether this spatial clustering also correlates with some particular functional output, we characterized the differentially expressed genes they harbor. Using IMR90 as an example, TiLO TAD clusters not marked by HMGB1 and HMGB1-hotspot ones contain a large fraction of all genes differentially expressed upon senescence entry

(i.e., in total ~50% of up- and > 20% of downregulated genes). However, non-HMGB1 clusters uniquely harbor genes involved in replication, RNA localization, and p53 signaling, where HMGB1-hotspot clusters are notably rich in genes relevant to the SASP and proinflammatory signaling (Fig 3D). Of the 313 upregulated genes in these hotspot clusters, > 60 were linked to SASP production and 46 were not only upregulated in senescent IMR90 but also discovered as secreted SASP factors in fibroblasts via proteomics (Basisty *et al*, 2020). More than half of these genes directly relied on HMGB1 loss for induction (Fig 3E). This also held true when HUVEC HMGB1 hotspot TiLO clusters were analyzed (Appendix Fig S4G). In summary, TiLO has the power to identify spatial co-associations of TADs, explaining the functional specialization of different genomic domains demarcated by HMGB1.

### HMGB1 depletion underlies induction of the senescence program

It was previously shown that transduction of WI-38 human fibroblasts with shRNAs against *HMGB1* suffices for senescence induction (Davalos *et al*, 2013). Here, we treated HUVECs with self-delivering siRNAs targeting *HMGB1*. This led to a ~ 2-fold reduction in HMGB1 protein and RNA levels within 72 h (Appendix Fig S5A) accompanied by a doubling of β-galactosidase and p21-positive cells in knockdown populations, but by only small changes in nuclear size and BrdU incorporation (Appendix Fig S5B–E). To obtain stronger effects, we turned to IMR90 where standard siRNA transfections allowed for a > 10-fold decrease in HMGB1 protein and RNA levels, as well as to changing expression of such senescence markers as *CDKN1B* and *HMGA1* (without affecting *HMGB2*; Fig 4A). Analysis of RNA-seq data from siRNA-treated and control IMR90 returned ~ 900 up- and > 950 downregulated genes upon *HMGB1* knockdown (Fig 4B). GO term and gene set enrichment analyses showed that the upregulated genes could be linked to proinflammatory signaling, while downregulated ones associated with changes in chromatin organization, transcriptional silencing, and the p53 pathway, all hallmarks of senescence entry (Fig 4C and D). Looking for direct HMGB1 targets in knockdown data, we identified 104 up- and 121 downregulated genes bound by HMGB1; they showed mean ChIP-seq signal enrichment at their 5' or 3' ends. Reassuringly, upregulated genes were linked to NF-κB and p38 signaling, as well as exocytosis; downregulated ones were linked to growth signaling, chromatin reorganization, and cell cycle arrest (Appendix Fig S5F). Interestingly, comparison of significant

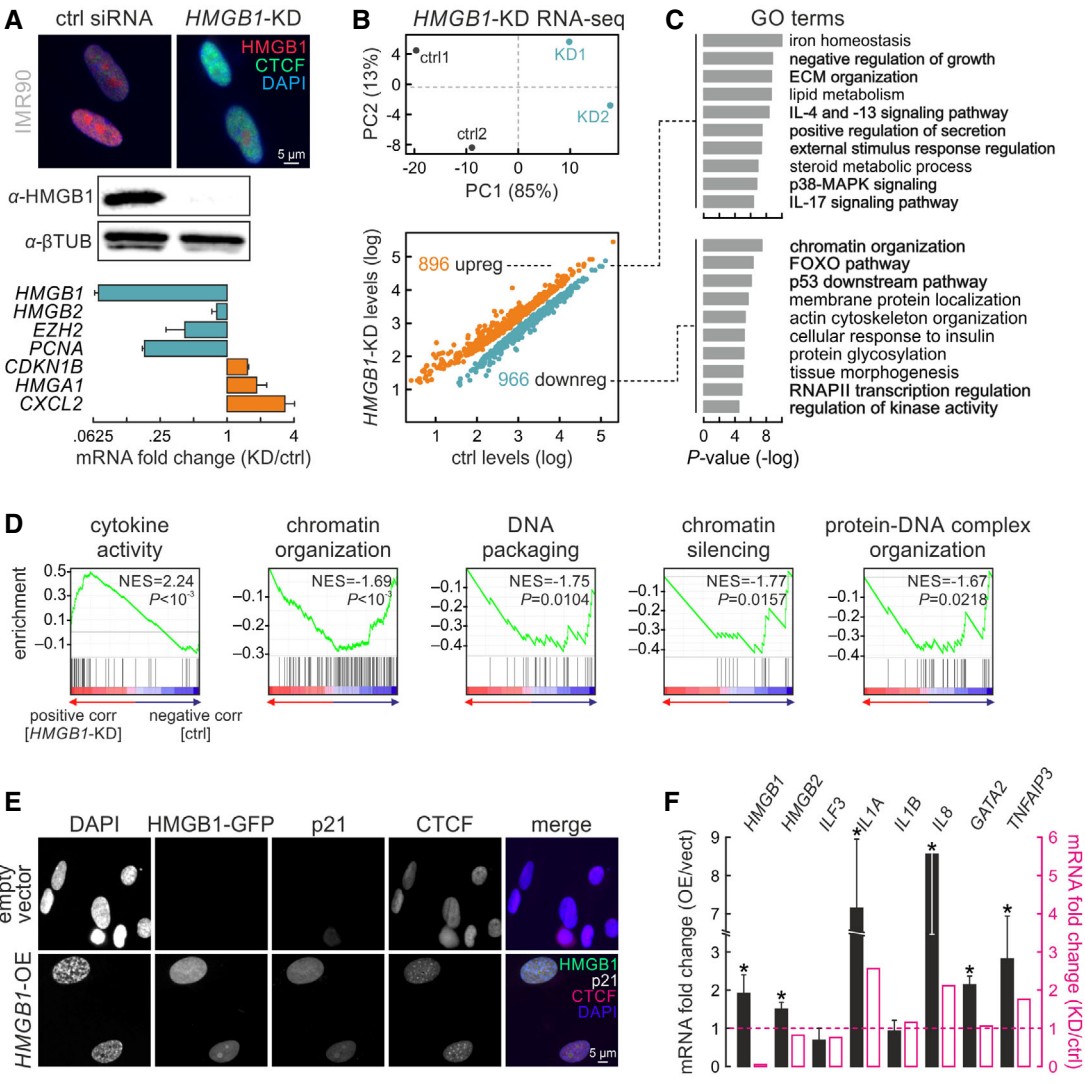

**Figure 4. Modulating HMGB1 expression induces senescence-specific gene expression changes.**

A  Immunofluorescence (*top*), Western blot (*middle*), and RT–qPCR analyses (*bottom*; mean fold change ± SD from two biological replicates) confirm *HMGB1* knockdown in IMR90.

B  PCA analysis plot (*top*) of control (*black*) and *HMGB1* knockdown replicates (*blue*). Scatter plot (*bottom*) showing significantly up- (> 0.6 log₂-fold change; *orange*) or downregulated genes (< −0.6 log₂-fold change; *green*) upon *HMGB1* knockdown.

C  Bar plots showing GO terms associated with the up/downregulated genes from panel B and their enrichment *P*-values (−log; *right*).

D  Gene set enrichment analysis of *HMGB1*-KD data. Normalized enrichment scores (NES) and associated *P*-values for each gene set are shown.

E  Representative images of IMR90 overexpressing HMGB1-GFP, immunostained for p21 and CTCF, and counterstained with DAPI. IMR90 transfected with empty vectors provide a control. Bar: 5 μm.

F  Bar plots showing RT–qPCR data (mean mRNA fold change ± SD, from two biological replicates) for selected genes in *HMGB1*-overexpressing compared to control IMR90. The mean fold change for each mRNA from *HMGB1* knockdown RNA-seq data is also shown for comparison (*magenta bars*). *P < 0.01; unpaired two-tailed Student's *t*-test.

gene expression changes upon *HMGB1* knockdown to those from senescence entry cells revealed poor correlation (Appendix Fig S5G). Genes downregulated in both knockdown and senescent cells were notably linked to RNA splicing, processing, and cleavage. Genes downregulated in senescence and upregulated upon *HMGB1* knockdown were relevant to RNA metabolism, but also the stress response and apoptosis (Appendix Fig S5G). Thus, HMGB1 appears to control a specific leg of the senescence gene expression program.

We complemented our knockdown experiments with *HMGB1* overexpression. We expressed an HMGB1-GFP fusion protein in IMR90 using a doxycycline-inducible PiggyBac system (Fig 4E). We selected for transfected cells using antibiotics, but refrained from generating single cell-derived populations to gauge heterogeneity arising from differences in integration sites. Within < 10 h of over-expression induction, nuclear accumulation of HMGB1 in a subset of the population led to a strong increase in p21 signal, to the emergence of characteristic DAPI-dense foci and, in some cases, to the

formation of senescence-induced CTCF clusters (Zirkel *et al*, 2018; Fig 4E). All these are hallmarks of senescence entry and agree with changes in SASP gene expression following *HMGB1* overexpression (Fig 4F). Thus, increased HMGB1 levels can also drive senescence entry, most likely via reinforced paracrine signaling.

### HMGB1 binds and regulates a discrete set of senescence-related mRNAs

The number of protein-coding loci bound by HMGB1 and regulated upon senescence entry and *HMGB1* knockdown does not explain the full extent of the senescence program (Appendix Figs S2F and G, and S5F). To address this disparity, we pursued the idea that HMGB1 also acts as an RNA-binding protein (RBP), as was suggested by recent classifications of the human proteome (Castello *et al*, 2016; Trendel *et al*, 2019). This idea was reinforced by our analysis of Ribo- and RNA-seq data (Appendix Figs S1F and S5G) and by our cataloguing of HMGB1 protein partners in proliferating IMR90. Mass spectrometry revealed a broad range of RBPs and splicing regulators co-immunoprecipitating with HMGB1, in addition to the expected chromatin-associated proteins (Fig 5A) and despite coIPs being performed after RNase A treatment of samples to avoid indirect interactions. In the end, ~40% of HMGB1 interactors qualify as RBPs (Fig 5B).

To study HMGB1 as an RNA binder, we applied sCLIP (Kargapolova *et al*, 2017) to proliferating IMR90 (Appendix Fig S6A and B). Analysis of two well-correlated replicates (Appendix Fig S6C) provided a set of 1,773 binding peaks on 866 different transcripts (Fig 5C). Reassuringly, HMGB1-bound mRNAs display < 14% overlap to HMGB1-bound genes in ChIP-seq (Appendix Fig S6D). Thus, cross-linking of HMGB1 to RNA cannot simply be the by-product of its binding to transcriptionally active chromatin loci.

On RNA, HMGB1 mostly binds exons and 5'/3' UTRs, but also a substantial number of non-coding RNAs (Fig 5C and D). HMGB1-bound sites present the same hexameric 5'-NMWGRA-3' (M = A/C, W = A/T, R = A/G) motif irrespective of the predicted folding of the underlying RNA (Fig 5C and Appendix Fig S6E). Much like what we observed in ChIP-seq, HMGB1 binds ~3-fold more transcripts that are up- rather than downregulated upon senescence. Upregulated mRNAs associated with senescence-related GO terms such as ECM organization, wound healing, and negative regulation of cell

proliferation, while downregulated ones mostly with processes such as RNA splicing, RNA-/miRNA-mediated gene silencing or histone remodeling and deacetylation (Appendix Fig S6F). After crossing sCLIP with RNA-seq data from *HMGB1* knockdown IMR90, 56 up- and 97 downregulated mRNAs were found bound by HMGB1. Curiously, upregulated transcripts showed a slight bias for HMGB1 binding in their 5' ends, while downregulated ones showed stronger 3' end binding (Appendix Fig S6G and H). Consistent with all previous observations, upregulated mRNAs could be linked to processes such as ECM organization and inflammatory activation, while downregulated mRNAs to splicing and chromatin reorganization (Appendix Fig S6H).

As RBPs are known to affect the splicing patterns of their target transcripts, we examined how splicing is altered upon senescence entry by IMR90 using Whippet (Sterne-Weiler *et al*, 2018). We documented ~ 4,000 significant changes in mRNA splicing, the majority of which concerned alternative usage of transcription start and polyadenylation sites (> 80% of cases; Fig 5E), consistent with recent observations in senescing HUVECs (Shen *et al*, 2019). This trend remained essentially invariable when we interrogated the kind of splicing changes occurring upon *HMGB1* knockdown or to HMGB1-bound and differentially spliced mRNAs (Fig 5E). A large fraction (~ 26%) of splicing events in mRNAs bound by HMGB1 in sCLIP overlap events seen upon both senescent and HMGB1-KD IMR90 (and > 95% overlap the events seen in at least one of the two conditions; Fig 5F). Differentially spliced mRNAs shared by senescence entry and *HMGB1* knockdown encode factors linked to such senescence-regulated processes as cell cycle and cell growth regulation, and the p53 pathway (Fig 5G), while those additionally bound by HMGB1 show a bias toward processes linked to RNA splicing and biogenesis (Fig 5H). Thus, the nuclear loss of HMGB1 correlates with changes to the cell's transcriptome processing.

In light of this unforeseen role, we revisited the HMGB1 interactome (Fig 5A) and found that 15 HMGB1 protein partners were downregulated in senescence, but also upon *HMGB1* knockdown. Of these, 12 qualified as RBPs (Castello *et al*, 2016). BCLAF1 (Shao *et al*, 2016), ILF3 (Tominaga-Yamanaka *et al*, 2012; Wu *et al*, 2015), PTBP1 (Georgilis *et al*, 2018), RAN (Cekan *et al*, 2016; Gu *et al*, 2016; Sobuz *et al*, 2019), SRSF7 (Chen *et al*, 2017), and TRA2B (Chen *et al*, 2018) were recently implicated in senescence and SASP regulation (Fig 5I, *starred*), while ZC3H18 was identified as a mediator of

---

**Figure 5. HMGB1 binds specific mRNAs and its loss affects splicing.**

A   Volcano plot (*left*) showing mass-spec data for proteins co-immunoprecipitating with HMGB1. Statistically enriched HMGB1 interactors (*orange dots*) associate with the GO terms/pathways illustrated in the network analysis (*right*; node size reflects the number of proteins it includes; proteins are listed in Dataset EV4).

B   Venn diagram showing 1/3 of HMGB1 interactors classifying as RNA-binding proteins (according to Castello *et al*, 2016).

C   Genome browser views showing HMGB1 sCLIP data (*black*) along the *ASH1L*, *CCNL2*, and *TET2* loci; input tracks (*gray*) provide background levels. *: significantly enriched peaks. The consensus motif for HMGB1 binding on RNA is also shown (*bottom right*).

D   Bar graphs showing genomic distribution of HMGB1 RNA-bound peaks ($\log_2$ enrichment).

E   Bar plots showing relative occurrence of differential-splicing events in IMR90 undergoing senescence (*left*), in HMGB1 knockdown IMR90 (*middle*), or in HMGB1-bound mRNAs (*right*). The number of bound mRNAs (*N*) analyzed is indicated below each bar.

F   Venn diagram showing differential-splicing events shared between conditions from panel (F).

G   Heatmaps showing GO terms/pathways associated with differentially spliced mRNAs shared between senescence entry and HMGB1 knockdown from panel (F).

H   As in panel G, but for the 75 mRNAs also bound by HMGB1 in sCLIP data.

I   Venn diagram (*left*) showing 15 HMGB1 interacting proteins from panel (A) are also downregulated upon both senescence entry and *HMGB1* knockdown in IMR90. Of these, 12 are RBPs, 6 have been implicated in senescence (*asterisks*), and 4 are bound by HMGB1 in sCLIP data (*arrows*).

J   Bar graphs showing mean fold change in selected mRNAs (over ± SD from two biological replicates) from *ILF3* (*purple*), *RBMX* (*white*), or *PNN* knockdown experiments (*blue*) in proliferating IMR90. *: significantly different to siRNA controls; *P* < 0.01, unpaired two-tailed Student's *t*-test. Dashed line indicates no change in expression.

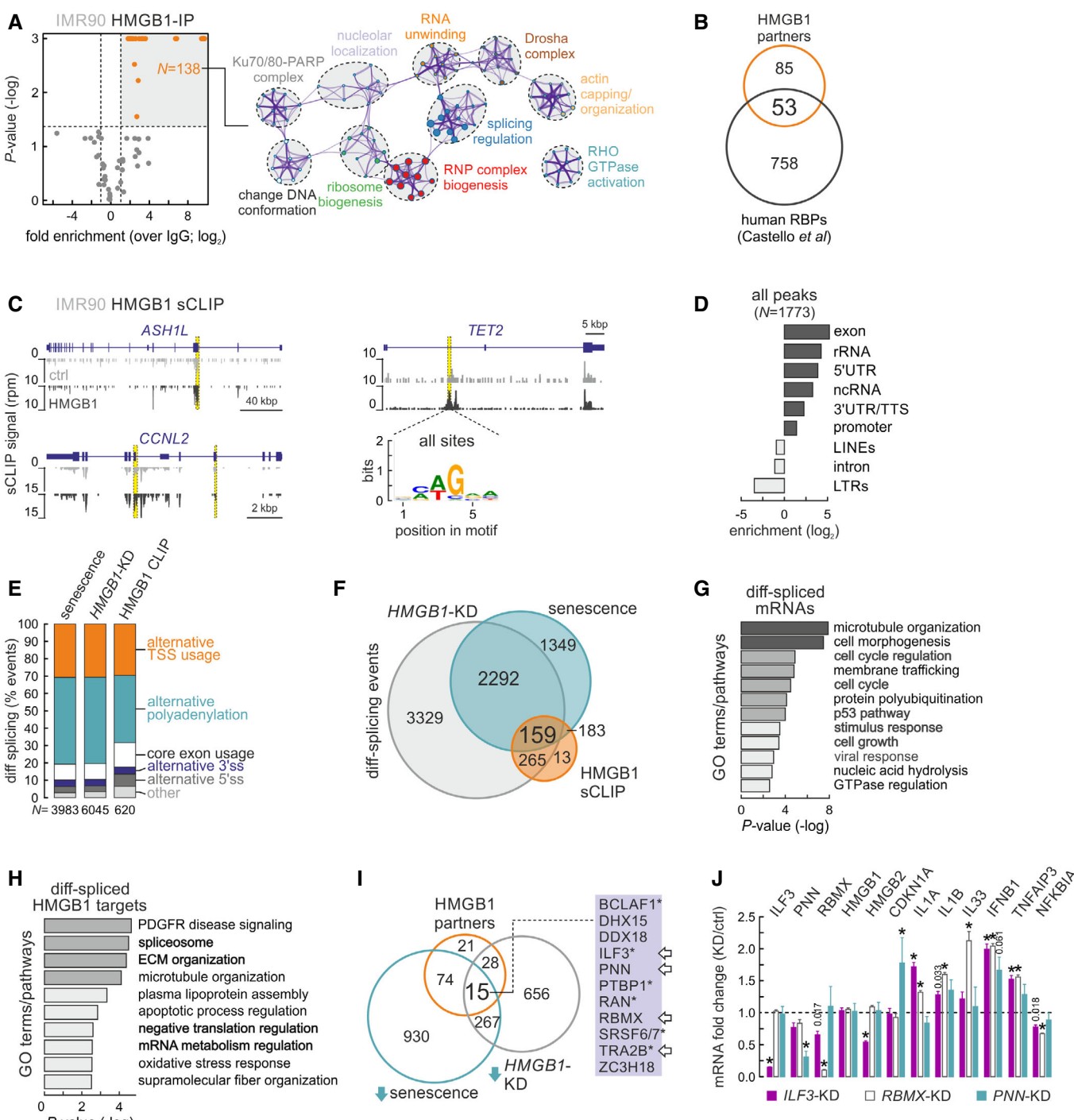

Figure 5.

NF-κB activation (Gewurz et al, 2012). Strikingly, the mRNAs of *ILF3*, *PNN*, *RBMX*, and *TRA2B* are also directly bound by HMGB1 in sCLIP data (Fig 5I, *arrows*). Query of the STRING database (https://string-db.org; Szklarczyk et al, 2019) links all these RBPs in a single network relevant to RNA processing (Appendix Fig S6I).

To validate this, we first performed co-immunoprecipitation experiments showing that HMGB1 and ILF3 physically interact (Appendix Fig S6J). We next showed that, although ILF3 is down-regulated upon *HMGB1* knockdown, its protein levels increase upon HMGB1 overexpression. This increase also manifests in a paracrine manner, since cells not carrying the HMGB1 overexpression cassette still show elevated ILF3 (Appendix Fig S6K). In addition, the ILF3 increase coincides with NF-κB translocation into cell nuclei signifying inflammatory activation of cells (Appendix Fig S6L). In line with what was reported for oncogene-induced senescence (Tominaga-Yamanaka et al, 2012), ILF3 binds SASP-relevant mRNAs in proliferating IMR90 (Appendix Fig S6M) and its senescence-induced loss (Appendix Fig S6N) can lead to their stabilization for translation.

Notably, in our model, ILF3 also binds HMGB2 mRNA (Appendix Fig S6M) and may thus be implicated in a feedforward regulatory loop for this factor. Finally, we knocked down *ILF3*, *RBMX*, and *PNN* individually in proliferating IMR90. With the exception of p21 induction, *PNN* knockdown did not affect *HMGB1/2* or SASP-related mRNA levels. In contrast, *ILF3* and *RBMX* knockdown led to the upregulation of interleukins, *IFNB1* or *TNFAIP3* (Fig 5J), all related to inflammatory activation. Interestingly, *HMGB1* levels did not change in any of these experimental setups, while *ILF3*-knockdown did suppress *HMGB2*. On this basis, we infer that HMGB1 is an upstream regulator of this cascade (given its binding on the mRNAs of all three RBPs; Fig 5I), while ILF3 can specifically modulate HMGB2 (which cannot happen via *HMGB1* knockdown; Fig 4A). Thus, HMGB1 is central to a regulatory circuit comprising RBP cofactors that regulate one another, as well as cell-autonomous and paracrine senescence.

# Discussion

Unlike the well-documented extracellular role of HMGB1 as a proinflammatory stimulus, its positioning along mammalian chromosomes and the transcriptional control it exerts are poorly understood. Here, we assign a multifaceted role to HMGB1—first as an on-chromatin regulator of active gene loci, and then as a *bona fide* RNA-binding protein regulating a distinct subset of mRNAs. Together, we deduce that HMGB1 acts to "buffer" gene expression levels, its loss from senescent cell nuclei mostly triggering upregulation of target loci and mRNAs. In addition, a subset of HMGB1-bound positions mark TAD/loop domain boundaries. Domains rich in HMGB1 binding peaks preferentially co-associate in 3D nuclear space and harbor SASP-related genes induced upon both senescence entry and *HMGB1* knockdown. This suggests that HMGB1 loss also affects chromatin topology in a manner relevant to gene expression changes. Interestingly, 3D chromatin domains rich in HMGB1 are generally depleted of CTCF loops. This implies that these spatial conformations may be incompatible and form by different mechanisms.

In its novel role as a direct RNA-binding regulator, HMGB1 is part of RNP complexes that affect transcript splicing and processing. In fact, a number of HMGB1 partners are RBPs implicated in the regulation of senescence induction and the SASP (Tominaga-Yamanaka *et al*, 2012; Wu *et al*, 2015; Cekan *et al*, 2016; Gu *et al*, 2016; Shao *et al*, 2016; Chen *et al*, 2017; Georgilis *et al*, 2018; Sobuz *et al*, 2019). In addition, the mRNAs of four of these RBPs (i.e., *ILF3*, *PNN*, *RBMX*, and *TRA2B*) are also direct HMGB1 targets. This renders HMGB1 a central player also in RNA homeostasis. Once HMGB1 is depleted from nuclei, mRNAs necessary for paracrine senescence can become stabilized (Tominaga-Yamanaka *et al*, 2012). This constitutes a remarkable example of a regulatory circuit, where deregulation of genes, transcripts, and topology in one cellular compartment (the nucleus) is in direct and quantitative crosstalk with signaling deployed in another (the extracellular space in which HMGB1 is secreted). Our observations come to substantiate previous hypotheses of low nuclear HMGB1 titers being necessary for the full deployment of the SASP (Davalos *et al*, 2013). Thus, the cascade regulating senescence entry has a strong, almost hierarchical, dependency on the nuclear events preceding SASP induction.

Recently, we characterized the function of the sister protein to HMGB1, HMGB2, as regards entry into replicative senescence (Zirkel *et al*, 2018). The loss of HMGB2 appears to precede that of HMGB1 and drives formation of prominent senescence-induced CTCF clusters (SICCs). This affects the spatial architecture of chromosomes and concomitantly gene expression. Intriguingly, HMGB2 target loci are also usually upregulated once relieved of HMGB2 binding; however, this is the only similarity between the functions of HMGB1 and HMGB2. The loss of HMGB1 does not trigger SICC formation, the same way that the loss of HMGB2 does not trigger p21 activation or SASP induction. Also, HMGB1 and HMGB2 bind non-overlapping genomic loci and demarcate TADs in distinct modes—HMGB2 marks the extremities of TADs that shift one boundary upon senescence entry, while HMGB1 is mostly found at invariable TAD/loop domain boundaries enriched for SASP-related genes. Critically, knockdown of *HMGB1* does not reduce *HMGB2* levels in our cells. Conversely, knocking down *HMGB2* does not affect *HMGB1* levels (Zirkel *et al*, 2018), meaning that the pathways these two factors control do not overlap, but are rather deployed in parallel. However, the nuclear loss of either HMGB coincides with changes in lamins, cohesin, and CTCF, which will also contribute to the effects observed in senescence.

Finally, *HMGB1* knockdown in primary lung fibroblasts leads to gene expression changes that are partially inversed upon senescence entry of the same cells (e.g., the negative regulation of RNAPII transcription is suppressed in the knockdown, but not in senescence; *MYC* activation is upregulated in the knockdown, but suppressed upon senescence entry). This may be interpreted as a coordinated counter-regulation of HMGB1-driven effects on the path to senescence and can be explained by the fact that the nuclear presence of HMGB1 is linked to favorable proautophagic effects that enhance cell survival and limit programmed cell death (Tang *et al*, 2010). This might also be a simple way to explain the strong overexpression of *HMGB1* in various cancer types (Tang *et al*, 2010; Li *et al*, 2014). This overexpression, although highly deleterious for normal cells, seems to favor increased cell proliferation (Kang *et al*, 2013; Li *et al*, 2014). Thus, the nuclear abundance of HMGB1 (and likely also of HMGB2) can be seen as a marker for proliferative capacity: Senescent cells have essentially no nuclear HMGBs, while continuously dividing cancer cells display levels even higher than those seen in normal tissue. In a next step, deciphering the functional implications behind this "readout" may potentially help us understand how a subset of cells escape senescence to acquire a malignant identity.

# Materials and Methods

### Primary cell culture and senescence markers

Human umbilical vein endothelial cells from single, apparently healthy, donors (passages 2–3; Lonza) were continuously passaged to replicative exhaustion in complete Endopan-2 supplemented with 2% FBS under 5% $CO_2$. Cells were constantly seeded at ~ 10,000 cells/cm², except for late passages when they were seeded at ~ 20,000 cells/cm². Single IMR90 isolates (I90-10 and I90-79, passage 5; Coriell Biorepository) were continuously passaged to replicative exhaustion in MEM (M4655, Sigma-Aldrich)

supplemented with non-essential amino acids and 10% FBS under 5% $CO_2$. Senescence-associated β-galactosidase assay (Cell Signaling) was performed according to the manufacturer's instructions to evaluate the fraction of positively stained cells at different passages. Cell proliferation was monitored by MTT assays at different passages. In brief, ~ 5,000 cells are seeded into 96-well format plates in quadruplicates. On the next day, the medium is replaced with 100 ml fresh medium plus 10 ml of a 12 mM MTT stock solution (Invitrogen), and cells are incubated at 37°C for 4 h. Subsequently, all but 25 ml of the medium is removed from the wells, and formazan dissolved in 50 ml DMSO, mixed thoroughly and incubated at 37°C for 10 min. Samples are then mixed again and absorbance read at 530 nm. Measurements are taken at 24, 48, and 72 h post-seeding, background subtracted, and normalized to the 24-h time point. Finally, nascent DNA synthesis was monitored by EdU incorporation and subsequent labeling with Alexa 488 Fluors (Click-iT EdU Imaging Kit; Invitrogen). In brief, cells were incubated in 10 mM EdU for 7 h, fixed using 3.7% PFA/PBS for 15 min at room temperature, permeabilized, and labeled as per manufacturer's instructions, before imaging on a widefield Leica microscope.

## Immunofluorescence and image analysis

Proliferating and senescent cells were grown on coverslips from the stage indicated and were fixed in 4% PFA/PBS for 15 min at room temperature. After washing once in PBS, cells were permeabilized in 0.5% Triton-X/PBS for 5 min at room temperature. Blocking with 1% BSA/PBS for 1h was followed by incubation with the following primary antibodies for 1–2 h at the indicated dilution: mouse monoclonal anti-HMGB1 (1:1,000; Abcam ab190377-1F3); rabbit polyclonal anti-HMGB2 (1:1,000; Abcam ab67282); mouse monoclonal anti-HMGB1/2 (1:1,000; Sigma-Aldrich 12248-3D2); rabbit polyclonal anti-CTCF (1:500; Active motif 61311); rabbit polyclonal anti-H3K27me3 (1:1,000; Diagenode C15410069); mouse monoclonal anti-p21 (1:500; Abcam ab184640-GT1032); rabbit polyclonal anti-lamin B1 (1:2,000; Abcam ab16048); and mouse monoclonal anti-β-tubulin (1:1,000; Sigma-Aldrich T0198-D66). Following immunodetection, cells were washed twice with PBS for 5 min before incubating with secondary antibodies for 1 h at room temperature. Nuclei were stained with DAPI (Sigma-Aldrich) for 5 min, washed, and coverslips mounted onto slides in Prolong Gold Antifade (Invitrogen). Note that for gSTED microscopy only, the 2C Pack STED 775 secondary antibodies (1:2,000; Abberior 2-0032-052-6) were used. For image acquisition, a widefield Leica DMI 6000B with an HCX PL APO 63x/1.40 (Oil) objective was used; confocal and super-resolution images were acquired on a Leica TCS SP8 gSTED microscope with a 100×/1.40 (Oil) STED Orange objective. For immunofluorescence image analysis, the NuclearParticleDetector2D of the MiToBo plugin (ver. 1.4.3; available at http://mitobo.informa tik.uni-halle.de/index.php/Main_Page) was used. Measurements of nuclear immunofluorescence signal were automatically generated using a mask drawn on DAPI staining to define nuclear bounds. Background subtractions were then implemented to precisely determine the mean intensity per area of each immunodetected protein. Deconvolution of super-resolution images was performed using the default settings of the Huygens software (Scientific Volume Imaging).

## Whole-cell protein extraction, Western blotting, and mass spectrometry

For assessing protein abundance at different passages, ~4 × 10^6 cells were gently scraped off 15-cm dishes and pelleted for 5 min at 600 g. The supernatant was discarded, and the pellet resuspended in 100 ml of ice-cold RIPA lysis buffer (20 mM Tris–HCl pH 7.5, 150 mM NaCl, 1 mM EDTA pH 8.0, 1 mM EGTA pH 8.0, 1% NP-40, 1% sodium deoxycholate) containing 1× protease inhibitor cocktail (Roche), incubated for 20 min on ice, and centrifuged for 15 min at > 20,000 g to pellet cell debris and collect the supernatant. The concentration of nuclear extracts was determined using the Pierce BCA Protein Assay Kit (Thermo Fisher Scientific), before extracts were aliquoted and stored at −70°C to be used for Western blotting. For fractionations, the protocol previously described was used (Watrin et al, 2006) with addition of 1.5 µM EGS (10 min, room temperature) to stabilize HMGB1 on chromatin. Resolved proteins were detected using the antisera mentioned above, plus a mouse monoclonal anti-H3K9me3 (1:200; Active motif 39286). For whole-cell proteomics, extracts in RIPA buffer were analyzed by the CECAD proteomic core facility in biological triplicates on a Q-Exactive Plus Orbitrap platform (Thermo Scientific) coupled to an EASY nLc 1000 UPLC system with column lengths of up to 50 cm.

## HMGB1 sCLIP and analysis

sCLIP was performed on ~ 25 million UV-cross-linked nuclei from proliferating IMR90 as previously described (Kargapolova et al, 2017) using the same the monoclonal HMGB1 antiserum (DSHB; PCRP-HMGB1-4F10) as for ChIP. Following sequencing of strand-specific libraries on a HiSeq4000 platform (Illumina), raw reads were mapped to the human reference genome (hg19). Consistent peaks were identified by overlapping intervals of peaks with a P-value < 0.05 from 2 biological replicates to obtain 1,773 peaks. This peak annotation was used to count reads uniquely aligned to each peak region using HTSeq, HMGB1-bound transcript coordinates were retrieved via Ensembl (GRCh37) and annotated using HOMER (http://homer.ucsd.edu), and Gene Ontology analysis was performed using Metascape (www.metascape.org). Finally, the final merged peak list was used for de novo motif analysis using ssHMM (Heller et al, 2017) and significantly enriched motifs were compared with existing RBP motifs via Tomtom (http://meme-suite.org/tools/tomtom). HMGB1-bound mRNAs are listed in Dataset EV1.

## Chromatin immunoprecipitation (ChIP) sequencing and analysis

For each batch of ChIP experiments, ~ 25 million proliferating cells, cultured to > 80% confluence in 15-cm dishes, were cross-linked in 1.5 mM EGS/PBS (ethylene-glycol-bis-succinimidyl-succinate; Thermo) for 20 min at room temperature, followed by fixation for 40 min at 4°C in 1%PFA. From this point onward, cells were processed via the ChIP-IT High Sensitivity kit (Active motif) as per manufacturer's instructions. In brief, chromatin was sheared to 200–500 bp fragments on a Bioruptor Plus (Diagenode; 2× 9 cycles of 30-s on and 30-s off at the highest power setting), and immunoprecipitation was carried out by adding 4 µg of a monoclonal HMGB1 antiserum (Developmental Studies Hybridoma Bank; PCRP-HMGB1-4F10) to ~ 30 µg of chromatin and rotating overnight at 4°C

in the presence of protease inhibitors. Following addition of protein A/G agarose beads and washing, DNA was purified using the ChIP DNA Clean & Concentrator Kit (Zymo Research) and used in next-generation sequencing on a HiSeq4000 platform (Illumina) to obtain at least 25 million reads were obtained of both sample and its respective ''input''. Raw reads (typically 100 bp-long) were processed with Encode ChIP-Seq pipeline (v1.5.1; ENCODE Project Consortium, 2012). Reads were mapped to hg19 human reference genome with BWA (Li, 2013); subsequent processing and filtering steps were performed with default pipeline settings by Picard (https://broadinstitute.github.io/picard/), BEDTools (Quinlan & Hall, 2010), Phantompeaktools, and SPP peak callers. Aligned reads were used to produce IP/Input signal coverage tracks with MACS2 (Zhang et al, 2008). IDR thresholded HMGB1 ChIP-seq peaks (conservative peak calling) per each cell type were annotated using Chipseeker (Yu et al, 2015) and are listed in Dataset EV2; Signal coverage files representing log2-fold change in IP/Input signal were used in Deeptools (v3.5.0, Ramírez et al, 2016) for plotting signal coverage over particular genomic positions for different conditions/cell types. Finally, transcription factor recognition motif enrichments within DHS footprints under HMGB1 ChIP-seq peaks were calculated using the Regulatory Genomics Toolbox (Gusmao et al, 2014). Note that all other ChIP-seq datasets used here come from previous work (Zirkel et al, 2018), with the exception of publicly available CTCF ChiP-Seq datasets for IMR90 (SRR639078) and HUVEC (ENCSR000ALA), RAD21 for IMR90 (ENCSR000EFJ), H3K4me3 for IMR90 (ENCSR431UUY) and HUVEC (ENCFF203KHF), and H3K27ac for IMR90 (ENCSR002YRE) and HUVEC (ENCFF 038HNR). IMR90 CTCF ChIP-seq was processed with Encode ChiP-Seq pipeline to match the rest of used datasets.

## Total RNA isolation, sequencing, and analysis

Control and HMGB1 knockdown were harvested in TRIzol LS (Life Technologies), and total RNA was isolated and DNase I-treated using the DirectZol RNA miniprep kit (Zymo Research). Following selection on poly(dT) beads, barcoded cDNA libraries were generated using the TruSeq RNA library Kit (Illumina) and were paired-end sequenced to at least 50 million read pairs on a HiSeq4000 platform (Illumina). Raw reads were mapped to the human reference genome (hg19) using default settings of the STAR aligner (Dobin et al, 2013), followed by quantification of unique counts using featureCounts (Liao et al, 2014). Counts were further normalized via the RUVs function of RUVseq (Risso et al, 2014) to estimate factors of unwanted variation using those genes in the replicates for which the covariates of interest remain constant and correct for unwanted variation, before differential gene expression was estimated using DESeq2 (Love et al, 2014). Genes with an FDR $< 0.01$ and an absolute ($\log_2$) fold change of $> 0.6$ were deemed as differentially expressed and listed in Dataset EV3. For splicing analysis, a reference index on the basis of hg19 annotation was first constructed, combined with all splice sites contained in the mapped RNA-seq reads. Raw reads were then aligned using Whippet (Sterne-Weiler et al, 2018) to the constructed index in order to quantify and annotate alternative splicing events. Subsequent plots were plotted using BoxPlotR (http://shiny.chemgrid.org/boxplotr/) and GO term enrichment bar plots using Metascape (http://metascape. org/gp/index.html; Zhou et al, 2019).

## siRNA-mediated HMGB1 knockdown and overexpression

Human umbilical vein endothelial cells were seeded at $\sim 20,000$ cells/cm$^2$ the day before transfection. Self-delivering Accell-siRNA pools (Dharmacon) targeting HMGB1, plus a non-targeting control (NTC; fluorescently tagged to allow transfection efficiency to be monitored), were added to the cells at a final concentration of 1 mM. Knockdown efficiency was assessed 72 h after transfection using RT–qPCR and immunofluorescence. For IMR90 cells, transfections using two different siRNAs and RNAiMAX (Invitrogen) were carried out as previously described (Zirkel et al, 2018). For HMGB1-GFP overexpression using the PiggyBac system, details were exactly as described previously for HMGB2 (Zirkel et al, 2018) with the difference that cells only tolerated a < 10-h induction of overexpression via doxycycline. In brief, the HMGB1 open reading frame was subcloned into the doxycycline-inducible KA0717 expression vector to generate an HMGB1-GFP fusion. The construct was co-transfected into IMR90 together with transactivator and transposase-encoding vectors (KA0637 and SBI Biosciences #PB200PA-1, respectively) at a DNA mass ratio of 10:1:3 using Fugene HD (Promega) as per manufacturer's instructions. Stable, transgene-positive, proliferating IMR90 were selected using 250 mg/ ml G418 (Sigma-Aldrich) and driven into senescence before being either harvested in TRIzol for downstream RT–qPCR analysis or cross-linked with 4% paraformaldehyde for 5 min at room temperature for immunofluorescence analyses.

## RNA immunoprecipitation (RIP) experiments

Approximately $5 \times 10^7$ proliferating (p. 16; for HMGB1 and ILF3 IP) and senescent IMR90 (p. 35; for HMGB1 IP) were scraped on ice in PBS (Sigma), pelleted by centrifugation, and lysed in ice-cold Polysome lysis buffer (100 mM KCl, 5 mM MgCl$_2$, 10 mM HEPES-NaOH pH 7.0, 1 mM DTT, 200 U/ml RNAseIn, 1x PIC, 0.5% NP-40) for 30 min. To ensure complete cell lysis, lysates were homogenized via 10 strokes with a Dounce homogenizer, passed through 27½-gauge needle 4 times, and sonicated for $2 \times 6$ cycles (30-s on/30-s off, low input) on a Bioruptor sonicator (Diagenode). Lysates were next treated with 400 U/ml DNase I (Worthington, LS006343) for 30 min and centrifuged at maximum speed for 10 min to collect supernatants. 5% of the cell lysate was saved as input, and the rest was subjected to overnight immunoprecipitation at 4°C in the presence of 10 μg HMGB1 (DSHB, 4F10; 68 μg/μl), ILF3 (ABclonal Biotechnology, A2496), or IgG antibody (Millipore, 12-371B; 1 μg/ μl). Next day, 30 μl of protein-G beads (Dynabeads, Invitrogen) per IP was pre-washed 3 times with 1 ml of NT2 buffer (x buffer contains 250 mM Tris–HCl pH 7.4, 750 mM NaCl, 5 mM MgCl$_2$, 0.25% NP-40) and incubated for 1 h at 4°C with 25 μl/IP of mouse bridging antibody (Active Motif; 1 μg/μl) with end-to-end rotation. After three washes with NT2 buffer, the beads were added to the lysates and incubated at 4°C for 2 h under end-to-end rotation. After incubation, beads were washed 6 times with 1 ml of NT2 buffer for 3 min under end-to-end rotation between washes. Finally, samples were resuspended in TRIzol (Invitrogen) and RNA was purified using the Direct-zol RNA Miniprep Kit (Zymo). Reverse transcription was carried out using the SuperScript™ II Reverse Transcriptase according to manufacturer's instructions (Invitrogen) and qPCRs using the qPCRBIO SyGreen Mix Separate-ROX (NIPPON).

## Co-immunoprecipitation coupled to mass spectrometry

Approx. $6 \times 10^6$ proliferating IMR90s were gently scraped and pelleted for 5 min at 600 $g$, supernatant discarded, and the pellet resuspended in 500 µl of ice-cold lysis buffer (20 mM Tris–HCl pH 8.0, 1% NP-40, 150 mM NaCl, 2 mM EDTA pH 8.0) supplemented with 1× protease inhibitor cocktail (Roche). This mixture was then incubated for 20 min on ice, followed by three cycles of sonication (30 s on, 30 s off, low input) and RNase A treatment, before centrifugation for 15 min at > 20,000 $g$ to pellet cell debris and collect the supernatant. While lysates were precleared, 30 µl protein-G magnetic beads (active motif) and 10 µg of HMGB1 anti-serum (PCRP-HMGB14F10s; DSHB) were incubated for 3 h at 4°C under rotation. Subsequently, the beads were captured on a magnetic rack (active motif) and added to the lysates for incubation at 4°C overnight under rotation. Next day, the beads were captured, washed four times with 800 µl ice-cold wash buffer I (50 mM Tris, 0,05% NP-40, and 50 mM NaCl), two times with 500 µl of wash buffer II (150 mM NaCl, 50 Mm Tris), recaptured, supernatant discarded, and purified proteins were predigested in 50 µl elution buffer (2 M urea, 50 mM Tris pH 7.5, 1 mM DTT, 50 ng trypsin) for 30 min at room temperature with gentle agitation. Following addition of 50 µl digestion buffer (2 M Urea dissolved in 50 mM Tris pH 7.5 and 5 mM chloroacetamide) and incubation for 30 min, another 50 µl of elution buffer supplemented with 50 ng of LysC and 100 ng of trypsin were added to each tube. Proteins were digested overnight at room temperature, the digestion was stopped by adding 1 µl tri-fluoroacetic acid, and peptides of each experiment were split in half, purified on two C18 stage tips, and all three replicates were analyzed by the CECAD proteomic core facility as above (all results are detailed in Dataset EV4).

## Ribo-seq and analysis

High-throughput ribosome profiling (Ribo-seq) on proliferating and senescent IMR90 was performed in collaboration with Ribomaps Ltd (https://ribomaps.com) according to an established protocol (Ivanov et al, 2018). Three independent replicas of proliferating or senescent IMR90 were grown, harvested in ice-cold polysome isolation buffer supplemented with cycloheximide, and shipped to Ribomaps for further processing and library preparation. Approx. 15% of each lysate was kept for isolation of RNA and used for RNA-seq of poly(A)-enriched fractions on a HiSeq2500 platform (Illumina). Following sequencing of both Ribo- and mRNA-seq libraries, the per base sequencing quality of each replicate passed the quality threshold, raw read counts were assigned to each protein-coding open reading frame (CDS) for Ribo-seq and to each transcript for mRNA-seq, and replicate correlations were tested. Read length distribution for Ribo-seq datasets fell within the expected range of 25–35 nt, with a peak between 28 and 32 nt showing strong periodic signals and an enrichment in annotated CDSs. For mRNA-seq, read lengths ranged between 47 and 51 nt and distributed uniformly across transcripts. For differential gene expression analysis, anota2seq (Oertlin et al, 2019) was used. Changes in Ribo-seq data represent changes in the ribosome occupancy of the annotated protein-coding open reading frame (CDS), and thus, only ribosome-protected fragments that map to the CDS were used in the analysis. VST normalized counts outputted using DESeq2 (Love et al, 2014) and inputted into

anota2seq were used for all subsequent downstream analysis. Differences in genes that pass a default false discovery rate (FDR) threshold of 15% were considered regulated. Such significant differences are then categorized into one of the following three modes: (i) translational: Changes in Ribo-seq that are not explained by changes in RNA-seq and imply changes at the protein level are due to changes at the translational level; (ii) mRNA abundance: Matching changes in RNA-Seq and Ribo-Seq that infer changes at the protein level are predominantly induced by changes at the transcriptional level; (iii) buffering: changes in RNA-seq that are not explained by changes in Ribo-seq and suggest maintenance of constant protein levels induced by changes at the transcriptional level or vice versa. Detailed results per gene locus and condition are listed in Dataset EV5.

## Whole-genome chromosome conformation capture (Hi-C) and TiLO analysis

Hi-C data from proliferating and senescent HUVEC and IMR90 were generated previously (Zirkel et al, 2018), and the HiTC Bioconductor package was used to annotate, correct data for biases in genomic features (Servant et al, 2012), and visualize 2D heatmaps with a maximum resolution of 25 kbp at which TADs were also called via TADtool (Kruse et al, 2016). For plotting insulation and "loop-o-gram" heatmaps, normalized interaction values in the twenty 25-kbp bins around each HMGB1 peak were added up, normalized to the median value in each matrix, and plotted provided the local maxima are higher than the third quantile of Hi-C data in the matrix. All R scripts were described previously (Zirkel et al, 2018). Loop calling was performed with FAN-C (Kruse et al, 2020) set of loops calling and filtering commands at resolution of 20 kbp.

For Topologically intrinsic Lexicographic Ordering (TiLO), we directly applied an algorithm from mathematical knot theory that makes zero assumptions about the structure, shape, or number of clusters in the data (Johnson, 2012). In brief, topologically intrinsic ordering was used to permutate the linear order of TADs (as the starting organization level in the Hi-C matrices) until a certain "robustly irreducible" topological condition is satisfied. Then, the "pinch ratio" algorithm (Heisterkamp & Johnson, 2013) was applied to heuristically slice the network at connections between TADs exhibiting local interaction minima, while also considering noise in the matrices. Finally, this analysis returns a list of TADs grouped into multiple clusters in cis, also via its built-in measure for network robustness defining the end-point. For rendering 3D chromosome model of IMR90 and HUVEC TiLO data, the Chrom3D interface was used (Paulsen et al, 2018). For all chromosomes per cell type, intra-TAD interactions were specified according to TILO output (Dataset EV6). Association with LADs was added as described in the Chrom3D manual for each chromosome (https://github.com/Chrom3D). LADs for proliferating and senescent IMR90 cells were inferred from LMNB1 ChIP-seq data (GSM1197635). Reads were aligned to the hg19 reference genome using Bowtie 2 v2.3 with default parameters and merged using SAMtools v1.9. The outputs were applied to EDD v1.0 for LAD calling also using default parameters (Lund et al, 2014). In the absence of similar data for HUVECs, constitutive LAD positions (cLADs and ciLADs) were downloaded from the LAD atlas (Meuleman et al, 2013) and used the same way. In the end, a .gtrack file (Chrom3D input) for chromosome visualization was produced

using Chrom3D scripts (https://github.com/Chrom3D/preprocess_scripts). Next, a *.BED* file specifying the genomic positions of the TADs (1 TAD = 1 bead) was created, and any gaps between them were filled as described in the Chrom3D manual. TADs belonging to the same contiguous TiLO cluster were grouped, and a separate *.gtrack* file was created for each TILO cluster containing only intra-chromosomal interactions. All single-base beads corresponding to gaps between TADs were removed from the final file. Finally, *.gtrack* files corresponding to each cluster were merged and inputted in Chrom3D, using 200,000 iterations (-n), a nuclear radius of 5 (-r), and a scale total volume of the beads relative to the volume of the nucleus set to 0.15 (-y). For whole genome visualizations that take into account interchromosomal interactions, Hi-C data were analyzed via HiCPro v2.11.4 at 40-kbp and 1-Mbp resolution, before LADs, TADs, and Hi-C matrices were used for the production of a diploid *.gtrack* file using default parameters; chromosomes Y and M were removed. IDs of beads containing HMGB1 peaks were identified and colored using the script *processing_scripts/color_beads.py* and the *blend* keyword to maintain coloring. The script for TiLO input preparation is provided as Dataset EV7.

**Statistical tests**

*P*-values associated with Student's *t*-tests and Fischer's exact tests were calculated using GraphPad (https://graphpad.com/), those associated with the Wilcoxon–Mann–Whitney test using R, and those with the hypergeometric test using an online tool (http://nemates.org/MA/progs/overlap_stats.html). Unless otherwise stated, *P*-values < 0.01 were deemed as statistically significant.

# Data availability

All NGS data generated in this study have been deposited to the NCBI Gene Expression Omnibus (GEO) repository as part of the GSE171782 SuperSeries as follows

- Hi-C and RNA-seq from proliferating/senescent HUVEC and IMR90 under accession number GSE98448 [https://www.ncbi.nlm.nih.gov/geo/query/acc.cgi?acc = GSE98448]
- sCLIP from proliferating IMR90 under accession number GSE146047 [https://www.ncbi.nlm.nih.gov/geo/query/acc.cgi?acc = GSE146047]
- Ribo-seq data from proliferating/senescent IMR90 under accession number GSE171780 [https://www.ncbi.nlm.nih.gov/geo/query/acc.cgi?acc = GSE171780]
- HMGB1 ChIP-seq data from proliferating/senescent HUVEC and IMR90 under accession number GSE171781 [https://www.ncbi.nlm.nih.gov/geo/query/acc.cgi?acc = GSE171781]
- HMGB1 knockdown RNA-seq data from IMR90 under accession number GSE171779 [https://www.ncbi.nlm.nih.gov/geo/query/acc.cgi?acc = GSE171779]

**Expanded View** for this article is available online.

## Acknowledgements

We would like to thank members of all laboratories involved in this study for helpful discussions, Leo Kurian for feedback on this manuscript, and Jonas Paulsen for help with Chrom3D. We thank the CMMC FACS sorting and CECAD Proteomics facilities for assistance. Work in the lab of AP was supported by CMMC Junior Research Group core funding, by the German Research Foundation (DFG) via TRR81 (Project 109546710), SPP2202 Priority program (Project 422389065) and a Basic module grant (Project 285697699), as well as by an Else-Kroener-Fresenius-Stiftung grant (Project 2015_A125). KS, NJ, and NÜ were also supported by the International Max Planck Research School for Genome Science, part of the GAUSS/GGNB, JK by the Irish Research Council (Project EPSPD/2019/214), IP by Erasmus+ Mobility funds, and YK by the TRR259 (Project 397484323). We also acknowledge support by the Open Access Publication Funds of the Göttingen University. Open Access funding enabled and organized by Projekt DEAL.

## Author contributions

KS, NJ, YK, NÜ, AZ, IP, TG, and AM performed experiments; GL, JK, and AM generated and analyzed Ribo-seq data; NJ, MN, YK, IV, and EGG performed computational analyses; CB and JA performed all NGS; AP conceived the study and wrote the manuscript with input from all coauthors.

## Conflict of interest

The authors declare that they have no conflict of interest.

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
