## [Review Process File · Molecular Systems Biology]

HMGB1 coordinates SASP-related chromatin folding and RNA homeostasis on the path to senescence

Konstantinos Sofiadis, Natasa Josipovic, Milos Nikolic, Yulia Kargapolova, Nadine Übelmesser, Iliana Varamogianni, Anne Zirkel, Ioanna Papadionysiou, Gary Loughran, James Keane, Audrey Michel, Eduardo Gusmao, Christian Becker, Janine Altmueller, Theodore Georgomanolis, Athanasia Mizi, and Argyris Papantonis

DOI: [10.15252/msb.20209760](https://doi.org/10.15252/msb.20209760)

Corresponding author(s): Argyris Papantonis (argyris.papantonis@med.uni-goettingen.de)

Review Timeline:

Submission Date:	30th May 20
Editorial Decision:	11th Jul 20
Revision Received:	5th Apr 21
Editorial Decision:	14th May 21
Revision Received:	17th May 21
Accepted:	19th May 21

Editor: Maria Polychronidou

Transaction Report:

Thank you again for submitting your work to Molecular Systems Biology. We have now heard back from the three referees who agreed to evaluate your study. Overall, the reviewers recognize that the presented findings on the dual role of HMGB1 seem potentially interesting. However, they raise a series of concerns, which we would ask you to address in a major revision.

The reviewers refer to the need to perform several additional analyses and to include further controls and replicates in order to strengthen the main conclusions. The reviewers' recommendations are rather clear and I think that they all provide constructive suggestions on how to improve the study. All issues raised by the reviewers would need to be convincingly addressed. Please let me know in case you would like to discuss any of the issues raised, I would be happy to schedule a call.

On a more editorial level, we would ask you to address the following issues.

REFEREE REPORTS

Reviewer #1:

In this paper, the authors used various methods to reveal the role of HMGB1 in senescence entry and senescence associated secretory phenotype (SASP). The links of HMGB1 with TADs and RNA splicing are interesting. However, the conclusion of HMGB1 affecting senescence need to be further addressed.

1. The authors showed that HMGB1 mRNA and protein both decreased in senescent cells. On top of that, the authors used knock-down of HMGB1 to address its biological function. However, it will make more sense to overexpress HMGB1 and then induce senescence to see whether the overdosed HMGB1 will impede the entry of senescence? If that is the case, it could be the reason why senescent cells tend to decrease HMGB1. Also, what if inducing senescence first, and then overexpress HMGB1, will senescent cells have less SASP after HMGB1 overexpression?

2. To confirm that HMGB1 has binding specificity to senescence associated genes, the comparison between HMGB1 knock-down and over-expressed cells is recommended. By doing ChIP-seq on both scenarios and examine the paralleled loss and gain of peaks will address the HMGB1 "binding specificity" in a more convincing way.
3. Page 7, line 1: "Reassuringly, HMGB1-bound mRNAs display <25% overlap to HMGB1-bound genes in ChIP-seq", where are the figures?
4. In page 6: "the population led to a strong increase in p21 signal", no supportive data could be found in Fig 5E or Fig 5F.
5. The senescence markers were mostly showed by seq data, such as GO pathway. This is weak and should add corresponding Western Blotting data to show the change of p16, p21, cell cycle proteins, NF-kB, phospho-p65, phosphor-ATM and SASP proteins.
6. How does HMGB1 affect the insulation? Does HMGB1 interact with any topological related proteins by doing Co-IP experiment?
7. Figure 1A, what does "D1" "D2" mean? Figure 1H, which is compared to which for the "UP/UP; UP/DOWN; DOWN/DOWN; DOWN/UP" comparison?
8. Figure 1B should include proliferating control cells.

Reviewer #2:

Manuscript Reference: MSB-20-9760

Sofiadis et al: HMGB1 coordinates SASP-related chromatin folding and RNA homeostasis on the path to senescence.

In senescence, the highly abundant protein HMGB1 translocates from the nucleus to the cytoplasm and is then secreted to trigger NFkB signalling. This is a key step in senescence and in particular in the senescence-associated secretory phenotype (SASP), but whether HMGB1 has additional roles in proliferating and senescent cells is poorly understood. This work uncovers a multifaceted role for HMGB1 in senescence. First, using a modified ChIP-seq protocol, the authors map HMGB1 binding on chromatin, and find that HMGB1 binding often coincides with the position of TAD boundaries (but is distinct from CTCF bound TAD boundaries), implicating HMGB1 in 3D genome architecture and gene expression control. Further, the authors find that HMGB1 also binds to RNAs where it affects both splicing and RNA homeostasis. This leads the authors to place HMGB1 in a central position within a regulatory hub that controls the entry into senescence.

This is a well written manuscript with technically sound data that support novel roles for HMGB1 in senescence. The dual role of HMGB1 described here as a modulator of 3D genome architecture and regulator at the level of RNA processing adds to its known role as proinflammatory extracellular stimulus, which is highly interesting and relevant for a broad readership. I have however some concerns which would need to be addressed before I could support publication, as outlined below.

Major points:

- 1.) Given that HMGB1 is extremely abundant (the most abundant non-histone protein in the nucleus), I find the number of ChIP-seq peaks (810 in HUVECs and 593 in IMR90s) surprisingly low. Does this mean that the majority of nuclear HMGB1 is not chromatin-bound?

2.) Have the authors actually shown that HMGB1 is displaced from its chromatin target sites (810 in HUVECs and 593 in IMR90s) in senescence? Is HMGB1 displaced from all these sites, or retained at some? If there are differences in HMGB1 chromatin binding between proliferating and senescent cells, how do these correlate with gene expression changes? I understand that HMGB1 is lost from nuclei upon entry into senescence, but how complete is the loss of the chromatin-bound fraction of HMGB1?

The authors set out to dissect the role of chromatin-bound HMGB1 in senescence - the question of how HMGB1 binding to chromatin changes in senescence appears to be central to this question and I don't think it has been addressed here.

3.) I am not sure the data shown fully supports the claim that HMGB1 peaks reside at TAD boundaries. For example, the line plot of HMGB1-marked boundaries in Figure 2E appears to indicate that the HMGB1 peaks are adjacent to TAD boundaries? And perhaps that there is a polarity of HMGB1 binding with regards to TAD boundaries too?

4.) Regarding the role of HMGB1 in RNA processing, I have a similar concern to the one I raised regarding the ChIP-seq data: I think this story is not complete without the HMGB1 sCLIP data in senescent cells. How much of the HMGB1 protein is retained bound to target RNAs in senescence?

5.) Related to concerns above: It is not clear to me how HMGB1 knock down and overexpression, respectively, affect the RNA- and chromatin-bound pool of HMGB1.

Minor points:

1.) Figure 1C: the four columns should be labelled in the figure. The legend indicates that passage 6 indicates the start and passages 21/32 the end point, but what are the two time points in between? Or are two time points for cells from two different donors shown?

2.) Figure 2H: I think the arrow in the bottom right panel (N=16) points in the wrong direction?

3.) "Significant and convergent changes between the two cell types involved strong suppression of histone chaperones and histones, lamin-associated proteins, centromere components, cohesin and condensing complexes, as well as all of HMGB/N-family proteins (Fig. 1B)."

I can't see data to support the claim that histone gene expression is repressed? In addition, I think researchers in the Polycomb field may object to EZH2 and PRC1 (is this Polycomb Repressive Complex 1 indeed? If so, which subunits?) be classified under 'heterochromatin'.

4.) Supplemental Figure 1C: The number of cells analyzed in each subgroup (N) is indicated. Have the exact same numbers of cells been analysed for HMGB1 and H3K27me3 staining, or is this a copy-paste error?

5.) "Comparative analysis of mRNA-seq and Ribo-seq data showed that essentially all significant changes at the level of mRNA translation were underlied by equivalent changes in transcript availability (Fig. 1F). Only few transcripts (~800) showed increased translation to counteract transcriptional suppression (e.g., TNFRSF19, HMG2, LMNB2) or the converse (e.g., IL12A, CDKN1A, CDKN2B)."

I am not questioning the overall conclusions, but 'essentially all' and 'few' should be substantiated by percentages. Also, picky comment but I am not sure 'were underlied' is grammatically correct.

6.) Curiously, several processes relevant to the SASP (e.g., NF- κ B signaling, mRNA metabolism) appeared to be regulated by combining higher transcription with diminished protein availability (Fig. 1H).

Parenthesis missing in this sentence, perhaps after 'metabolism'?

7.) Figure S2A and S2F: scale bars to indicate the size of the genomic windows shown are missing.

8.) "However, only few of these were strongly regulated upon senescence entry, with most being

upregulated (Fig. S2F, bottom).'
How many is 'only few'?

9.) Figure 4F legend y axis on the right: should this be 'mRNA fold change (KD/ctrl)' instead of 'mRNA fold change (KD/crtl)'

Reviewer #3:

In this study, Sofiadis et al. combine several data types to investigate the role of HMGB1 in the regulation of genes during replicative senescence. The key findings highlight a potential effect of HMGB1 in binding and regulating a small number of genes, binding to a subset of TAD borders and regulating some of the genes within those TADs. Interestingly, they also show evidence for a potential new role of HMGB1 in controlling RNA splicing. While the manuscript has some novelty: the RNA part, in particular, is potentially interesting, but overall the data robustness appears to be limited. For example, regarding the HMGB1 binding information (ChIP-seq), only one replicate is provided for each cell type.

Ribo-seq: Why is the Ribo-seq data analysed using different software than the mRNA-seq? Other Ribo-seq studies use very different analysis methods, usually coupling mRNA and ribosome profiles. Importantly, Ribo-seq samples are usually matched to the RNA-seq samples they are being compared to, in order to avoid batch effects confounding the comparison. How did the authors address this issue? I could not find the deposited Ribo-seq data or the proteomics? How does the Ribo-seq profile of control vs senescence compare to the findings of Loayza-Puch et al. 2013, Genome Biology? Perhaps they might want to cite this paper?

HMGB1 ChIP-seq: There seems to be only one replicate for each cell type (HUVEC and IMR90). It would be important to show multiple replicates to support any conclusion based on these data, especially when the authors claim that the two cell lines are very different - they share only 40 HMGB1 peaks as shown in Fig. S2C. HMGB1 ChIP-seq data in senescence conditions are also absent. Although HMGB1 is downregulated during senescence, it is not completely eradicated, as shown in this study.

Fig. 2E. HMGB1 at TAD boundaries: 21% of 810 peaks is a very small number to claim the association with TAD boundaries. The data seem quite noisy. Why is the HMGB1 boundary profile biased to the right side? The CTCF profile also does not appear as really centred on the boundary, with other comparable peaks to the left and right. I assume that the HMGB1 ChIP-seq signal in Fig. 2F, bottom, is from proliferating cells, aligned to senescence-TADs or is it actually HMGB1 ChIP-seq in senescent cells? Please clarify.

Fig. 2I There is no trend in 'sen-pro'? The number of loops in each plot would be worth mentioning.

Is the implementation of the TiLO algorithm used by the authors public, as this is a new application in this field? Did they try more widely used clustering algorithms like spectral clustering to see how the results vary? More analysis regarding the robustness of these results is needed.

Page7: "the nuclear loss of HMGB1 directly affects processing of the cell's transcriptome." The data are correlative and do not support this statement.

Fig. 4E: ectopic HMGB1 shows a nuclear localisation, but if the cells are senescent, don't we expect that HMGB1 (even if it is OE) to be released from the cells?

Minor points:

In all figures showing ChIP-seq profiles of HMGB1 and CTCF, why are the limits of the y-axis different? For example, in figure 2H (the bottom right is particularly low), keeping those the same might allow for a fairer comparison/assessment of HMGB1 binding at different locations.

I couldn't find the data deposited for the RNA-seq of the knock-down experiments?

On page 6 of the manuscript, several figure labels are mismatched: Fig. 5B, C, D etc. should be Fig. 4B,C,D.

Fig. 5A/C, the top of the figures seem to have a typo - IMR1-HMGB90 -> IMR90 - HMGB1?

Point-by-point response to reviewers' comments

Reviewer #1:

In this paper, the authors used various methods to reveal the role of HMGB1 in senescence entry and senescence associated secretory phenotype (SASP). The links of HMGB1 with TADs and RNA splicing are interesting. However, the conclusion of HMGB1 affecting senescence need to be further addressed.

We are happy to see that the reviewer found "the links of HMGB1 with TADs and RNA splicing interesting" and we provide below answers to all outstanding concerns.

1. The authors showed that HMGB1 mRNA and protein both decreased in senescent cells. On top of that, the authors used knock-down of HMGB1 to address its biological function. However, it will make more sense to overexpress HMGB1 and then induce senescence to see whether the overdosed HMGB1 will impede the entry of senescence? If that is the case, it could be the reason why senescent cells tend to decrease HMGB1. Also, what if inducing senescence first, and then overexpress HMGB1, will senescent cells have less SASP after HMGB1 overexpression?

We appreciate the suggestion, but unfortunately this experimental set up is unfeasible. Even very transient (e.g., >12 h) overexpression of HMGB1 in primary lines, like HUVECs and IMR90, is absolutely lethal for the cells. We experienced this using a PiggyBac vector carrying HMGB1-GFP, and it was also observed by the Campisi lab when using lenti-based overexpression (Davalos et al, *J Cell Biol*, 2014). As a result, there simply exists no large enough time-window to assess the impact of overexpression using genomics. However, we did manage to perform short-term overexpression (<10 h) followed by immunofluorescence that indicated senescence induction, rather than suppression (as judged by p21 upregulation; **Fig. 4E**), an effect similar to that inferred by Davalos *et al.*

2. To confirm that HMGB1 has binding specificity to senescence associated genes, the comparison between HMGB1 knock-down and over-expressed cells is recommended. By doing ChIP-seq on both scenarios and examine the paralleled loss and gain of peaks will address the HMGB1 "binding specificity" in a more convincing way.

As explained above, it is impossible to perform the overexpression experiment and obtain enough cells for the demanding HMGB1ChIP. At the same time, ChIP in HMGB1-knockdown cells would not yield any peaks given the absence of the factor (chromatin fractionation blots show <17% HMGB1 remaining after knockdown; **Appendix Fig. S5F**). Nevertheless, we went ahead and performed ChIP-seq in senescent IMR90 to find that the residual HMGB1 in senescent nuclei (~30% compared to normal HMGB1 levels; **Appendix Fig. S2C**) is not redirected to specific genomic targets (see **Fig. 2A-C**). On the basis of this new ChIP-seq data and replicates, we believe that specificity of our ChIP assays is validated (see also *de novo* HMGB1 binding motif convergence in **Appendix Fig. S2D**).

3. Page 7, line 1: "Reassuringly, HMGB1-bound mRNAs display <25% overlap to HMGB1-bound genes in ChIP-seq", where are the figures?

Since this was simply a Venn diagram, we only described it by text. However, we have now added this to **Appendix Fig. S2D** and the overlap is reduced to <15% due to our new ChIP-seq catalogues.

4. In page 6: "the population led to a strong increase in p21 signal", no supportive data could be found in Fig 5E or Fig 5F.

We apologize for the mistake; this was meant to refer to **Fig. 4E**; we have now corrected this.

5. The senescence markers were mostly showed by seq data, such as GO pathway. This is weak and should

add corresponding Western Blotting data to show the change of p16, p21, cell cycle proteins, NF- κ B, phospho-p65, phospho-ATM and SASP proteins.

We already show such data (direct protein levels changes using western blots or immunofluorescence) in **Figs 1C,E** and **4E**, but we have now added additional fractionation western blots in **Appendix Figs S1G, S2C, S5F, and S6K,L**, as well as whole-cell proteomics detailed in **Dataset EV4**. Note that we also corroborated many of our SASP-relevant genes on the basis of the recent SASP proteomics atlas (Basisty et al, *PLoS Biol*, 2020) in **Fig. 3E** and **Appendix Fig. S4G**.

6. How does HMGB1 affect the insulation? Does HMGB1 interact with any topological related proteins by doing Co-IP experiment?

We found no enrichment for known insulator/topological factors, like CTCF, cohesin/condensin subunits or ZNF143, in our HMGB1 coIP-mass spectrometry data (see **Dataset EV4** for details). Thus, we believe that the topological function of HMGB1 is most likely an effect of active chromatin-based contacts (already in the initial TAD discovery paper by Bing Ren's lab, Dixon et al, *Nature*, 2012, active promoters were found to demarcate a large proportion of TAD boundaries). We directly queried insulation score changes at HMGB1-marked TAD boundaries upon senescence entry and found diminished yet not abolished insulation (**Fig. 2G**). Notably, we now also found that HMGB1-rich TADs tend to cluster together in 3D nuclear space, as shown by new analysis combining TiLO and Chrom3D (Paulsen et al, *Nat Protoc*, 2018), and that domains rich in HMGB1-anchored interactions appear to be incompatible with CTCF loops in the same domain (see **Fig. 3A**). These features imply some independence of the HMGB1- and CTCF-based mechanisms, but more work will be needed in the future to fully understand this.

7. Figure 1A, what does "D1" "D2" mean? Figure 1H, which is compared to which for the "UP/UP; UP/DOWN; DOWN/DOWN; DOWN/UP" comparison?

The reviewer is correct in pointing out this oversight in **Fig. 1A**. We have worked with cells from two different donors for HUVECs (which we again named D1 and D2) and with two separate isolates of IMR90 (which we named D[onor] 1 and D[onor] 2). This was done to ensure uniformity to our HMGB2 work (Zirkel et al, *Mol Cell*, 2018). We now simplified this and explain details in the **Methods** section. Regarding **Fig. 1H**, it refers to proteome/transcriptome data comparisons shown in **Fig. 1G** (now clarified in the figure legend).

8. Figure 1B should include proliferating control cells.

Fig. 1B shows fold-change (\log_2) in expression comparing senescent to proliferating levels. We apologize for not making this clear in the first place. We have now amended the figure and legend to clarify this.

Reviewer #2:

In senescence, the highly abundant protein HMGB1 translocates from the nucleus to the cytoplasm and is then secreted to trigger NFκB signalling. This is a key step in senescence and in particular in the senescence-associated secretory phenotype (SASP), but whether HMGB1 has additional roles in proliferating and senescent cells is poorly understood. This work uncovers a multifaceted role for HMGB1 in senescence. First, using a modified ChIP-seq protocol, the authors map HMGB1 binding on chromatin, and find that HMGB1 binding often coincides with the position of TAD boundaries (but is distinct from CTCF bound TAD boundaries), implicating HMGB1 in 3D genome architecture and gene expression control. Further, the authors find that HMGB1 also binds to RNAs where it affects both splicing and RNA homeostasis. This leads the authors to place HMGB1 in a central position within a regulatory hub that controls the entry into senescence. This is a well written manuscript with technically sound data that support novel roles for HMGB1 in senescence. The dual role of HMGB1 described here as a modulator of 3D genome architecture and regulator at the level of RNA processing adds to its known role as proinflammatory extracellular stimulus, which is highly interesting and relevant for a broad readership. I have however some concerns which would need to be addressed before I could support publication, as outlined below.

We are glad to see that the reviewer finds our manuscript “well written”, “technically sound”, “relevant for a broad readership”, and in support of “novel roles for HMGB1 in senescence”.

Major points:

1.) Given that HMGB1 is extremely abundant (the most abundant non-histone protein in the nucleus), I find the number of ChIP-seq peaks (810 in HUVECs and 593 in IMR90s) surprisingly low. Does this mean that the majority of nuclear HMGB1 is not chromatin-bound?

This has long been also a concern of ours. We have been worried that we are underestimating the number of HMGB1 chromatin-bound positions, mostly due to our dual crosslinking approach that will inevitably generate higher “background noise” level in the data, and to the fact that although HMGBs are generally thought to be potent chromatin binders, they exhibit a highly dynamic association with chromatin that unfolds in a “stop-and-go” fashion in repeated short-lived cycles (reviewed in Stros, *Biophys Biochim Acta*, 2010 and in Reeves, *Biophys Biochim Acta*, 2010). To address this, we have now done the following. First, we performed replicates of HMGB1 ChIP-seq to increase statistical confidence in the data. Second, we reanalyzed all our ChIP-seq data not using our previous BWA/MACS2 pipeline, but rather the IDR-based pipeline introduced by the ENCODE consortium (v1.5.1; ENCODE Project Consortium, *Nature*, 2012). As a result, we can now identify approx. 2,000 HMGB1 peaks in proliferating IMR90 or HUVEC that paint a more comprehensive picture of the HMGB1 chromatin binding repertoire (see **Fig. 2** and **Appendix Fig. S2**). Finally, we present chromatin fractionation western blots that describe HMGB1 distribution in nucleoplasm and chromatin and how this changes upon senescence or HMGB1-knockdown (**Appendix Figs S2C and S5F**).

2.) Have the authors actually shown that HMGB1 is displaced from its chromatin target sites (810 in HUVECs and 593 in IMR90s) in senescence? Is HMGB1 displaced from all these sites, or retained at some? If there are differences in HMGB1 chromatin binding between proliferating and senescent cells, how do these correlate with gene expression changes? I understand that HMGB1 is lost from nuclei upon entry into senescence, but how complete is the loss of the chromatin-bound fraction of HMGB1? The authors set out to dissect the role of chromatin-bound HMGB1 in senescence - the question of how HMGB1 binding to chromatin changes in senescence appears to be central to this question and I don't think it has been addressed here.

This is a fair point. We have been reluctant to perform HMGB1 ChIP-seq in senescent cells for two reasons. First, because of the quantitative loss of the factor (see **Fig. 1C,E**). Second, due to our previous experience with HMGB2 (no peaks in senescence; Zirkel et al, *Mol Cell*, 2018). However, to address this comment by the reviewer, we performed HMGB1 ChIP-seq in senescent IMR90 (p. 34/35) in two independent replicates and were only able to retrieve ~40 peaks genome-wide. Thus, little HMGB1 remains focally bound to target

loci in senescence, and this is in line with fractionation western blots — all this new data is presented in revised **Fig. 2** and **Appendix Fig. S2**.

3.) I am not sure the data shown fully supports the claim that HMGB1 peaks reside at TAD boundaries. For example, the line plot of HMGB1-marked boundaries in Figure 2E appears to indicate that the HMGB1 peaks are adjacent to TAD boundaries? And perhaps that there is a polarity of HMGB1 binding with regards to TAD boundaries too?

This is a valid point. There are two issues to discuss here (and they have been the basis of this revision). First, that HMGB1 peaks designated as “residing at TAD boundaries” simply lie within a TAD-boundary Hi-C bin called at 40-kbp resolution. This, as the reviewer suggests, means that the HMGB1 peak might lie right adjacent to the position of strongest insulation. This of course holds true also for CTCF and any other factor designated as “boundary factor” give that ChIP-seq resolution is always higher than Hi-C resolution and one needs to assign peaks to “boundary” bins. A second, and more important, point is that upon doubling (or tripling for HUVECs) the number of HMGB1 peaks genome-wide, we still see a fraction of ~11% residing within TAD-boundary bins, but the overall signal distribution is now skewed in favor of intraTAD positions (see **Fig. 2E** and **Appendix Fig. S3B**). Nonetheless, we still identified robust boundaries depleted of CTCF and rich in HMGB1 (**Fig. 2F**). However, to fully remedy this issue, we also looked into “loop domains” at 20-kbp resolution, which is the highest our Hi-C data can afford. We found that ~1/3 of HMGB1 peaks mark the anchors of loop domains (**Fig. 2I** and **Appendix Fig. S3E**). Thus, we now extend our findings of topological demarcation by HMGB1 to higher resolution features nested within TADs that offer more confidence as regards HMGB1 positioning at sites of local insulation (see **Fig. 2G** for mean insulation scores at HMGB1 TAD boundaries). Finally, we could not find evidence of true polarity in HMGB1 boundary-binding.

4.) Regarding the role of HMGB1 in RNA processing, I have a similar concern to the one I raised regarding the ChIP-seq data: I think this story is not complete without the HMGB1 sCLIP data in senescent cells. How much of the HMGB1 protein is retained bound to target RNAs in senescence?

Again, we appreciate the concern and the suggestion. sCLIP is a very demanding protocol (also as regards availability of the factor of interest). Generating CLIP data from proliferating IMR90 was challenging enough, and we have repeatedly tried to do the same using senescent IMR90, but without success due to the low HMGB1 titers. To circumvent this, we performed an HMGB1 RIP experiment in proliferating and senescent cells and using RT-qPCR of multiple targets from our sCLIP data, we are now able to show that these are significantly less associated with HMGB1 upon senescence entry (see **Appendix Fig. S6N**).

5.) Related to concerns above: It is not clear to me how HMGB1 knock down and overexpression, respectively, affect the RNA- and chromatin-bound pool of HMGB1.

This is a valid, yet not feasible request. As we also explained in response to Rev. #1 (see above), HMGB1 overexpression is detrimental to both HUVECs and IMR90 and we are only able to sustain few cells for short-term (<10 h) overexpression via a PiggyBac vector. This means that it is impossible to harvest enough cells for ChIP/CLIP or even for fractionation western blots. On the other hand, performing such fractionation blots in HMGB1-knockdown cells is more feasible, and we see reduction down to 17% (**Appendix Fig. S5F**). The RNA-bound fraction reduction could be addressed by HMGB1 RIP-qPCR (**Appendix Fig. S6N**).

Minor points:

1.) Figure 1C: the four columns should be labelled in the figure. The legend indicates that passage 6 indicates the start and passages 21/32 the end point, but what are the two time points in between? Or are two time points for cells from two different donors shown?

All blots in **Fig. 1C** represent increasingly later passages of one HUVEC or IMR90 donor. We have now added intermediate passage information to this figure panel.

2.) *Figure 2H: I think the arrow in the bottom right panel (N=16) points in the wrong direction?*

We thank the reviewer for noticing this; we have now anyway replaced this panel.

3.) *"Significant and convergent changes between the two cell types involved strong suppression of histone chaperones and histones, lamin-associated proteins, centromere components, cohesin and condensing complexes, as well as all of HMGB/N-family proteins (Fig. 1B)."*

I can't see data to support the claim that histone gene expression is repressed? In addition, I think researchers in the Polycomb field may object to EZH2 and PRC1 (is this Polycomb Repressive Complex 1 indeed? If so, which subunits?) be classified under 'heterochromatin'.

H3K27me3-marked regions are generally classified as facultative heterochromatin (with H3K9me3-marked ones are classified as "constitutive heterochromatin"). Regarding PRC1, this was an inadvertent mistake, as this gene is not a Polycomb, but a cell cycle-related gene; it appears that senescence suppresses PRC2, but not PRC1 components. Last, as regards histone expression levels, we have removed that part of the heatmap but neglected to amend the text. We have now deleted the mention to "histones" (although this is a true observation already made by us and others).

4.) *Supplemental Figure 1C: The number of cells analyzed in each subgroup (N) is indicated. Have the exact same numbers of cells been analysed for HMGB1 and H3K27me3 staining, or is this a copy-paste error?*

The cells analysed were co-stained for HMGB1 and H3K27me3. We now clarify this in the figure legend.

5.) *"Comparative analysis of mRNA-seq and Ribo-seq data showed that essentially all significant changes at the level of mRNA translation were underlied by equivalent changes in transcript availability (Fig. 1F). Only few transcripts (~800) showed increased translation to counteract transcriptional suppression (e.g., TNFRSF19, HMG2, LMNB2) or the converse (e.g., IL12A, CDKN1A, CDKN2B)."*

I am not questioning the overall conclusions, but 'essentially all' and 'few' should be substantiated by percentages. Also, picky comment but I am not sure 'were underlied' is grammatically correct.

The reviewer is rightfully picky, "were underlied" is not correct and we rewrote this fragment to also include percentages for the numbers of transcripts mentioned.

6.) *Curiously, several processes relevant to the SASP (e.g., NF-κB signaling, mRNA metabolism appeared to be regulated by combining higher transcription with diminished protein availability (Fig. 1H).*

Parenthesis missing in this sentence, perhaps after 'metabolism'?

We thank the reviewer for noticing this. Indeed, a parenthesis was missing and is now added.

7.) *Figure S2A and S2F: scale bars to indicate the size of the genomic windows shown are missing.*

We have added scale bars to allow inference of the genomic window shown in all ChIP/CLIP panels.

8.) *"However, only few of these were strongly regulated upon senescence entry, with most being upregulated (Fig. S2F, bottom)."*

How many is 'only few'?

These were <10 genes, but we have replaced this panel altogether since "superenhancer" analysis did not reveal much that is of relevance to HMGB1 function in this context.

9.) *Figure 4F legend y axis on the right: should this be 'mRNA fold change (KD/ctrl)' instead of 'mRNA fold change (KD/crtl)'*

We are again glad the reviewer noticed this. It is now corrected.

Reviewer #3:

In this study, Sofiadis et al. combine several data types to investigate the role of HMGB1 in the regulation of genes during replicative senescence. The key findings highlight a potential effect of HMGB1 in binding and regulating a small number of genes, binding to a subset of TAD borders and regulating some of the genes within those TADs. Interestingly, they also show evidence for a potential new role of HMGB1 in controlling RNA splicing. While the manuscript has some novelty: the RNA part, in particular, is potentially interesting, but overall the data robustness appears to be limited. For example, regarding the HMGB1 binding information (ChIP-seq), only one replicate is provided for each cell type.

We appreciate the reviewer's acknowledgement of "novelty" in our work (especially as regards HMGB1's RBP role) that shows "evidence for a potential new role for HMGB1". We also understand the concern about ChIP-seq replicates, and have addressed this (see our answers below as well as to Reviewer #2 above).

Ribo-seq: Why is the Ribo-seq data analysed using different software than the mRNA-seq? Other Ribo-seq studies use very different analysis methods, usually coupling mRNA and ribosome profiles. Importantly, Ribo-seq samples are usually matched to the RNA-seq samples they are being compared to, in order to avoid batch effects confounding the comparison. How did the authors address this issue? I could not find the deposited Ribo-seq data or the proteomics? How does the Ribo-seq profile of control vs senescence compare to the findings of Loayza-Puch et al. 2013, Genome Biology? Perhaps they might want to cite this paper?

The reviewer is correct in stating that mRNA-seq and Ribo-seq need to be matched to avoid batch effects; this is exactly how we performed the experiment and analysis. We collected cells from three independent biological replicates at each time point in Ribo-seq lysis buffer, and provided 10% of the lysate for mRNA-seq and the rest for the Ribo-seq protocol (implemented by Ribomaps). Data analysis was also implemented by Ribomaps; they are a spin-off of the Baranov lab (University College Cork, Ireland), who are leading Ribo-seq experts (see Benitez-Carlos et al, *Genome Res*, 2020; Andreev et al, *eLife*, 2018; Yordanova et al, *Nature*, 2018; Andreev et al, *Genome Biol*, 2015), and this is the standard pipeline used by them. Nevertheless, we did perform a sanity check the moment this mRNA-seq data became available. We compared it to our RNA-seq data from IMR90 to find that these new datasets faithfully recapitulated the gene expression changes seen upon senescence entry. As a result, the correlation between the two analysis strategies is reassuringly high ($\rho > 0.8$ when comparing differentially-expressed genes in the two datasets).

We also thank the reviewer for pointing out the Loayza-Puch et al (*Genome Biol*, 2013) paper. We now refer to this data compared to ours, although the lack of replicative senescence as a model in this *Genome Biol* paper limits comparability. Finally, the full proteomics results are shown in **Dataset EV5** and raw and matching Ribo/mRNA-seq raw data can now be found in GEO under the SuperSeries GSE146047 (accessible using the following token: *sdojewambzaxzat*).

HMGB1 ChIP-seq: There seems to be only one replicate for each cell type (HUVEC and IMR90). It would be important to show multiple replicates to support any conclusion based on these data, especially when the authors claim that the two cell lines are very different - they share only 40 HMGB1 peaks as shown in Fig. S2C. HMGB1 ChIP-seq data in senescence conditions are also absent. Although HMGB1 is downregulated during senescence, it is not completely eradicated, as shown in this study.

In response to this concern, we have now replicated our ChIP-seq data, and have also performed HMGB1 ChIP-seq in senescent IMR90. As pointed out by the reviewer, inclusion of replicates increased the statistical confidence of our analysis, which is also now changed from the "traditional" BWA/MACS2 to the IDR-based pipeline used by the ENCODE consortium (v1.5.1; ENCODE Project Consortium, *Nature*, 2012). As a result, we now identify ~2,000 HMGB1 peaks in proliferating IMR90 or HUVEC, which offer a more comprehensive picture of HMGB1 chromatin binding repertoires (see new **Fig. 2** and **Appendix Fig. S2**). Still, HUVEC and IMR90 only share 579 peaks in the proliferating state. Now, as regards HMGB1 ChIP in senescence, we have

been reluctant to try this because of the quantitative loss of the factor (see **Fig. 1C,E**) and because of our previous experience with HMGB2 (no peaks in senescence; Zirkel et al, *Mol Cell*, 2018). Nevertheless, we performed HMGB1 ChIP-seq in senescent IMR90 (p. 34/35) in two replicates, and were only able to retrieve ~40 peaks genome-wide. Thus, little HMGB1 remains focally bound to target loci in senescence, and this is in line with fractionation blots — all this new data is presented in revised **Fig. 2** and **Appendix Fig. S2**.

Fig. 2E. HMGB1 at TAD boundaries: 21% of 810 peaks is a very small number to claim the association with TAD boundaries. The data seem quite noisy. Why is the HMGB1 boundary profile biased to the right side? The CTCF profile also does not appear as really centred on the boundary, with other comparable peaks to the left and right. I assume that the HMGB1 ChIP-seq signal in Fig. 2F, bottom, is from proliferating cells, aligned to senescence-TADs or is it actually HMGB1 ChIP-seq in senescent cells? Please clarify.

Given our now larger lists of HMGB1 peaks, we have redrawn all these figures. Upon doubling (or tripling in the case of HUVECs) the number of HMGB1 peaks genome-wide, we still see a fraction of ~11% residing within TAD-boundary bins, but the overall signal distribution is now clearly skewed in favor of intra-TAD positions (see **Figs 2E** and **Appendix Fig. S3B**). Nonetheless, we still identified boundaries depleted of CTCF and rich in HMGB1 (**Fig. 2F** and **Appendix Fig. S3C**). We acknowledge that the signal is noisy (mostly due to our dual crosslinking). To remedy this, we also looked into “loop domains” at 20-kbp resolution, which is the highest our Hi-C data can afford. We found that 30% of HMGB1 peaks actually mark IMR90 loop domain anchors (**Fig. 2I**). This extends our findings of topological demarcation by HMGB1 to higher resolution features nested within TADs and offer more confidence as regards HMGB1 positioning at sites of local insulation (see insulation score plots in **Fig. 2G**).

Fig. 2I There is no trend in 'sen-pro'? The number of loops in each plot would be worth mentioning.

The number of loops was shown under the bean plots at the bottom right of **Fig. 2I**. Nevertheless, we have replaced this panel in light of our new HMGB1 ChIP-seq data.

Is the implementation of the TiLO algorithm used by the authors public, as this is a new application in this field? Did they try more widely used clustering algorithms like spectral clustering to see how the results vary? More analysis regarding the robustness of these results is needed.

Yes, we provide the (rather simple) script for preparing input files for TiLO now as **Dataset EV7**. Then, TiLO is run via its public website: <http://www.cs.okstate.edu/~doug/src/prc/>. TiLO uses TAD intervals as a starting point (computed here at 40-kbp resolution using TADtool) and then estimates inter-TAD interactions within each chromosome in a manner similar to the “meta-TAD” analysis in Fraser et al, *Mol Syst Biol*, 2015. Although we did not try a spectral clustering approach, we did compare our TiLO data to “TAD cliques” derived using the Chrom3D approach (Paulsen et al, *Nat Protoc*, 2018), which also takes into account lamina and inter-chromosomal interactions — and we got similar results as regards TAD clustering (see **Fig. R1**).

Page7: "the nuclear loss of HMGB1 directly affects processing of the cell's transcriptome." The data are correlative and do not support this statement.

The reviewer is correct about this. We have now rephrased this sentence (and the tone of the whole section) to reflect the fact that HMGB1 binding to transcripts correlates with, rather than directly explains how, the transcriptome is affected upon senescence entry.

Fig. 4E: ectopic HMGB1 shows a nuclear localisation, but if the cells are senescent, don't we expect that HMGB1 (even if it is OE) to be released from the cells?

Apparently, the short-term induction (<10 h) and larger titers of HMGB1-GFP overwhelm the cells such that strong nuclear signal remains (as HMGB1 is continuously made). We did not test if HMGB1-GFP is released in the extracellular milieu via ELISA and cannot comment on this (imaging would not address this point).

Fig. R1 Comparison of TiLO-based or TAD clique-based Chrom3D models of IMR90 chr17. Visualisations of TAD clustering in chr17 from proliferating IMR90 using TiLO clusters (top row) or not (bottom row). TADs (spheres) are coloured by the TiLO cluster they belong to (left column) or according to their HMGB1 ChIP-seq content (middle column; grey – zero peaks, blue – 1 or 2 peaks, purple – 3 or 4 peaks, red – 5 or more peaks). Violin plots (right column) show 3D distances amongst TADs in each subgroup or between HMGB1-containing and non-containing TADs. *: significantly different, Wilcoxon-Mann-Whitney test.

Minor points:

In all figures showing ChIP-seq profiles of HMGB1 and CTCF, why are the limits of the y-axis different? For example, in figure 2H (the bottom right is particularly low), keeping those the same might allow for a fairer comparison/assessment of HMGB1 binding at different locations.

We agree with this point, and have now amended all such average graphs in the manuscript accordingly.

I couldn't find the data deposited for the RNA-seq of the knock-down experiments?

We sincerely apologize for this. Raw HMGB1-knockdown data can be again found on GEO under SuperSeries GSE146047 (accessible using the following token: *sdojewambzaxzat*). Also, all raw DEseq2 outputs are provided in **Dataset EV3**.

On page 6 of the manuscript, several figure labels are mismatched: Fig. 5B, C, D etc. should be Fig. 4B,C,D.

We apologize for the mistake, which we have now corrected.

Fig. 5A/C, the top of the figures seem to have a typo - IMR1-HMGB90 -> IMR90 - HMGB1?

We thank the reviewer for noticing this typo, which we have now corrected.

Thank you for sending us your revised manuscript. We have now heard back from the three reviewers who were asked to evaluate your study. Overall, the reviewers are satisfied with the performed revisions and are supportive of publication. As you will see below, reviewer #2 lists a few remaining concerns, which we would ask you to address in a minor revision.

On a more editorial level, we would ask you to address the following.

REFEREE REPORTS

Reviewer #1:

The biology behind SASP has acquired intense interests in senescence and aging field. While efforts proceed to reveal more observations, it's largely unknown how SASP is associated with the chromosomal spatial remodeling. This paper researched a highly abundant but drastically diminished protein upon senescence, HMGB1, together with its role in senescence entry and SASP regulation.

From the initial submission, the novelty and value of this paper have been recognized while several problems were requested for further address in order to consummate the soundness and strengthen the conclusion. By revision, the authors have optimized the HMGB1 CHIP in senescence,

which is an important piece of data. They also performed HMGB1 RIP experiment with good quality. My questions were also well addressed by the authors.

By compiling all the evidences from the authors, especially the correlation of ChIP-Seq, Hi-C and RIP-Seq, the involvement of HMGB1 in binding with certain topological boundaries on chromatin and with hundreds of mRNAs can be concluded. As a consequence, the loss of HMGB1 plays a specific role in mediating the senescence establishment. As a future direction, since how HMGB1 impacts on the senescence initiation and maintenance is still elusive, more mechanistical studies can be deployed.

In sum, I believe the quality of the revised manuscript is agreeable to be accepted for publication.

Reviewer #2:

The authors have addressed my concerns and suggestions adequately and I found the revised version of the manuscript improved. Despite its somewhat patchworky design (one could argue that the role of HMGB1 on chromatin and as an RNA binding protein merit two separate papers), I am convinced that this study provides novel and important insights into the role of HMGB1 in senescence. Reading the paper again, I have some additional minor questions and suggestions as detailed below. However, these should be fairly easy to address for the authors none of these should stand in the way of publication.

1.) CCL2 kDa size marker missing in Figure S1G

2.) HSC-70 loading control looks different in Figure S1G compared to Figure S2C, especially the 'proliferating soluble' column. Why is this?

3.) "The just 44 peaks discovered in our ChIP-seq replicates argue against this (Fig. 2A-C)." Please consider changing to "The fact that we discovered just 44 peaks in our ChIP-seq replicates argue against this possibility (Fig. 2A-C)." (Page 5).

4.) "Interestingly, HMGB1 consistently associated with more up- rather than downregulated in both HUVECs and IMR90." Page 5 - word missing, presumably 'genes'?

5.) "Of those, downregulated genes were involved cell cycle transitions, while upregulated ones to ECM organization, cell adhesion, and inflammatory signaling (Appendix Fig. S2G,H)." Please consider changing to "...while upregulated ones were related to..."? (page 5)

6.) "Together, this analysis demonstrates how HMGB1 binds active loci relevant for the induction of the senescence gene expression program." (page 5)
Demonstrates *how* or demonstrates *that* (I think it should be *that* as the insights into mechanisms (how) are rather indirect (TF binding sites)).

7.) Should 'shared' be in italics in Figure 3J Figure S3F?

8.) "Looking for direct HMGB1 targets in knockdown data, we identified 104 up- and 121 downregulated genes showing HMGB1 ChIP-seq signal enrichment at their 5' and 3' ends." (Page 7). Is it not possible that direct HMGB1 target genes have HMGB1 bound only at the 5' end (or

perhaps only at the 3' end)? Does HMGB1 have to bind on both ends? Please explain.

9.) "Thus, HMGB1 appears to control a specific leg of the senescence gene expression program." If that were true, I would expect the genes that are down-regulated upon HMGB1 knockdown to be a subset of the genes that are downregulated upon senescence entry. However there are also genes that are only down-regulated upon HMGB1 knockdown but not upon senescence entry. Is it therefore not more correct to state that HMGB1 controls some of the senescence genes but also a set of genes that are not related to senescence? Please clarify.

Reviewer #3:

The authors have addressed my questions satisfactorily.

The authors have made all requested editorial changes.

Thank you again for sending us your revised manuscript and for performing the requested changes. We are now satisfied with the modifications made and I am pleased to inform you that your paper has been accepted for publication.

Corresponding Author Name: Argyris Papantonis

Manuscript Number: MSB-20-9760